# CGSVD: Cascaded Granular Singular Value Decomposition for Large Language Model Compression

Yuli Chen [1][2]  Shuhao Zhang [2]  Jiale Han [3]  Fanshen Meng [2]  Haishen Jiang [2]  Bo Cheng [2]  Qiang Tong [1]  Xiulei Liu [1]

## Abstract

The exponential growth in the parameter scale of Large Language Models (LLMs) has precipitated an urgent demand for efficient compression techniques to facilitate practical deployment. To address this challenge, low-rank decomposition based on Singular Value Decomposition (SVD) offers a principled, hardware-friendly pathway for compressing LLMs without retraining. However, existing training-free approaches predominantly rely on uniform rank allocation, implicitly assuming homogeneous redundancy across the model depth and thereby neglecting the inherent non-uniformity of representational evolution. To bridge this gap, we introduce **CGSVD**, a **C**ascaded **G**ranular **S**ingular **V**alue **D**ecomposition framework that leverages a dual-level non-uniform allocation strategy to maximize semantic preservation. Specifically, we quantify inter-layer significance via angular distance and assess intra-layer compressibility through spectral entropy, enabling precise identification of critical architectural components. Furthermore, we propose an Iterative Residual Filling (IRF) mechanism to bridge the parameter gap caused by integer-rank truncation and ensure strict adherence to global compression targets. Extensive experiments on representative LLM families ranging from 3B to 13B parameters verify the superiority of our approach. Notably, under a 30% compression ratio on the LLaMA3.1-8B model, CGSVD achieves a remarkable average zero-shot accuracy boost of 6.08% and reduces perplexity by 15.46 compared to the baseline. We release the code[1] to facilitate future research.

## 1. Introduction

In recent years, Large Language Models (LLMs) have achieved notable progress in artificial intelligence, showing strong empirical performance on a broad range of generative and reasoning-intensive tasks (Li et al., 2026). However, these transformative abilities have been inextricably linked to a dramatic escalation in model size, presenting severe challenges regarding memory footprint and inference latency in resource-constrained deployment environments (Touvron et al., 2023a; Zhu et al., 2024). Recently, extensive research suggests that these over-parameterized models exhibit significant redundancy, with their weight matrices often residing on low-dimensional manifolds (Ren & Zhu, 2024; Bai et al., 2025). To exploit this intrinsic property, Singular Value Decomposition (SVD) (Denton et al., 2014) has been widely adopted to reparameterize dense layers into compact, low-rank forms, aiming to maximize representational efficiency while minimizing storage overhead.

SVD-based methods offer a mathematically principled and hardware-agnostic approach for achieving immediate inference acceleration, as they effectively bypass the intrinsic limitations associated with alternative compression techniques. Specifically, pruning methods often result in irregular weight sparsity that necessitates specialized hardware kernels to translate theoretical parameter reduction into actual latency gains (Frantar & Alistarh, 2023; Sun et al., 2024). Similarly, ultra-low-bit quantization introduces substantial discretization errors and rounding noise by forcing weights into extremely restricted numerical formats which can lead to significant loss in model fidelity (Xiao et al., 2023; Lin et al., 2024; Chen et al., 2025a). In contrast, SVD reorganizes weights into smaller dense matrices that remain natively compatible with standard linear algebra libraries while preserving the continuous nature of the weight

---

[1]Laboratory of Data Science and Information Studies, Beijing Information Science and Technology University, Beijing, China. [2]State Key Laboratory of Networking and Switching Technology, Beijing University of Posts and Telecommunications, Beijing, China. [3]Shenzhen Loop Area Institute, Shenzhen, China. Correspondence to: Bo Cheng <chengbo@bupt.edu.cn>, Xiulei Liu <liuxiulei@bistu.edu.cn>.

*Proceedings of the 43$^{rd}$ International Conference on Machine Learning*, Seoul, South Korea. PMLR 306, 2026. Copyright 2026 by the author(s).

---

[1]The code is available at: https://github.com/ironartisan/CGSVD.

distribution.

In the pursuit of higher reconstruction accuracy, recent research has branched into training-based frameworks such as ARS (Gao et al., 2024b) and Dobi-SVD (Wang et al., 2025b). These methods utilize differentiable rank selection or gradient-based refinement to recover performance. However, such approaches are often prohibitively time-consuming and computationally demanding as they require massive GPU resources and extensive optimization cycles. Current state-of-the-art training-free methods, such as ASVD (Yuan et al., 2023) and SVD-LLM (Wang et al., 2025c), introduce activation-aware whitening to prioritize features with high energy. Despite their efficacy, these methods typically rely on a uniform rank allocation strategy which implicitly assumes that all layers and functional modules within an LLM possess an identical degree of redundancy. This assumption is fundamentally suboptimal, as LLMs exhibit a pronounced U-shaped redundancy profile along their depth: shallow and deep layers are critical for representational grounding, whereas intermediate layers often exhibit strong linear dependency (Gromov et al., 2025). By failing to account for this non-uniformity, existing training-free techniques often suffer from catastrophic accuracy degradation at high compression ratios.

To address these limitations, we propose CGSVD, which is a high-efficiency framework designed to maximize semantic preservation through a non-uniform rank allocation strategy. As illustrated in Figure 1, CGSVD achieves granular precision by leveraging a three-stage decision pipeline. First, we project weights into an activation-aware manifold using Cholesky-based whitening. Second, we introduce a dual-level allocation mechanism where we quantify importance through angular distance at the inter-layer level and utilize spectral entropy at the intra-layer level to assess the intrinsic compressibility of specific projection modules. Finally, we develop an Iterative Residual Filling (IRF) strategy to eliminate the parameter waste inherent in integer-rank truncation by adaptively reallocating the remaining capacity to the most structurally critical modules.

We conduct extensive experiments on representative LLM families ranging from 3B to 13B parameters. Experimental results demonstrate that CGSVD consistently outperforms state-of-the-art training-free baselines.

Overall, our contributions are summarized as follows:

- We propose CGSVD, a training-free low-rank compression framework for LLMs that achieves substantial parameter reduction without retraining.

- We introduce a cascaded non-uniform rank allocation strategy that jointly models inter-layer importance via angular distance and intra-layer compressibility via spectral entropy.

- We develop an Iterative Residual Filling (IRF) mechanism to eliminate rank-induced parameter waste and ensure strict adherence to global compression targets.

- Evaluations across multiple LLM families from 3B to 13B parameters validate the robustness and scalability of CGSVD compared to prior SVD-based approaches.

## 2. Related Work

### 2.1. LLM Compression

The widespread deployment of LLMs is hindered by their prohibitive resource demands, prompting extensive research into pruning and quantization (Kim et al., 2025). Unstructured pruning methods, such as SparseGPT (Frantar & Alistarh, 2023) and Wanda (Sun et al., 2024), target individual redundant weights. While theoretically efficient, their practical utility is limited by the need for specialized hardware kernels to achieve inference acceleration. Conversely, structured pruning (Ma et al., 2023; Xia et al., 2024) offers a hardware-friendly alternative by removing entire components like heads or layers, yet this coarse granularity often disrupts neuronal coherence, causing irreversible accuracy loss. Parallel to pruning, quantization schemes (Frantar et al., 2022; Lin et al., 2024) reduce memory usage by lowering numerical precision, though they suffer from escalating discretization errors at ultra-low bit widths. Unlike these conventional methods, which retain original weight dimensions, SVD uniquely addresses these limitations by explicitly exploiting the intrinsic low-rank structure of neural layers. This fundamental property positions SVD as a mathematically rigorous foundation for compression, offering a viable path to balance model size and functional integrity.

### 2.2. SVD-based LLM Compression

SVD-based compression facilitates structured reduction by decomposing high-dimensional weight matrices into low-rank products, thereby transforming the computational topology and linearizing inference complexity (Liu et al., 2025). Early approaches focus on calibration to enhance reconstruction robustness. Methods such as FWSVD (Hsu et al., 2022) and ASVD (Yuan et al., 2023) weight singular values using Fisher information or activation statistics, effectively managing outlier distributions. While these methods improve upon standard decomposition, they often rely on uniform rank allocation, which neglects the heterogeneous sensitivity of different model layers. SVD-LLM (Wang et al., 2025c) advances this paradigm by incorporating a truncation-aware strategy that utilizes data whitening to minimize layer-wise errors, yet it still faces challenges in optimal rank distribution. To address the limitations of uniform rank assignment, AdaSVD (Li et al., 2025) optimizes

resource allocation through iterative updates, while ResSVD (Bai et al., 2025) further curbs error propagation by leveraging residual compensation and a partial-layer strategy. Beyond training-free methods, training-aware frameworks like ARS (Gao et al., 2024b) and Dobi-SVD (Wang et al., 2025b) attempt to optimize rank distribution through adaptive selection or differentiable learning. However, these learning-based techniques are computationally intensive and time-consuming, presenting substantial barriers to their practical application in large-scale LLM architectures.

## 3. Methodology

This section presents the proposed CGSVD framework, a training-free low-rank compression scheme designed to leverage the intrinsic statistical properties and layer-wise redundancies of Large Language Models (LLMs). The framework builds upon activation-aware weight reconstruction and employs a non-uniform rank allocation strategy to maximize the preservation of semantic expressive power under extremely low parameter budgets.

### 3.1. Problem Definition

Given a pretrained large language model $\mathcal{M}$, the core computations reside in a sequence of Transformer blocks, each containing $N$ high-dimensional linear projection layers $\{W_1, W_2, \ldots, W_N\}$. For any weight matrix $W \in \mathbb{R}^{m \times n}$, where $m$ and $n$ denote the output and input dimensions, respectively, the objective of low-rank compression is to approximate $W$ by decomposing it into the product of two low-rank matrices $A \in \mathbb{R}^{m \times r}$ and $B \in \mathbb{R}^{r \times n}$ such that the physical rank $r$ satisfies $r \ll \min(m, n)$. The total compressed parameter count is defined as $P_{comp} = \sum_{i=1}^{N} r_i(m_i + n_i)$. Our optimization goal is to minimize the performance loss, subject to a global target parameter retention ratio $\rho_{target}$:

$$\min_{\{r_i\}} \mathcal{L}(\mathcal{M}) \quad \text{s.t.} \quad \frac{\sum_{i=1}^{N} r_i(m_i + n_i)}{\sum_{i=1}^{N} m_i n_i} \leq \rho_{target} \quad (1)$$

where $\mathcal{L}$ denotes the loss function and $r_i$ represents the rank assigned to the $i$-th module. The key challenge lies in scientifically determining the optimal set of ranks $\{r_i\}$ based on the characteristics of various layers and modules.

### 3.2. Spatial Reconstruction via Activation Statistics

In LLMs, the importance of a weight matrix $W$ is highly dependent on the distribution of the input features $X$ (Yuan et al., 2023; Wang et al., 2025c). To capture this dynamic dependency, we first collect activation samples through a calibration dataset and compute the covariance matrix of the input features for each layer:

$$C_X = \mathbb{E}[X^T X] \in \mathbb{R}^{n \times n} \quad (2)$$

where $X$ represents the input tensor. The diagonal elements of $C_X$ reflect the activation intensity of different neurons, while the off-diagonal elements capture the correlations between features. To eliminate these correlations and homogenize the energy, we perform Cholesky decomposition (Lin, 2019) to extract a spatial reconstruction operator $L$:

$$C_X + \epsilon I = LL^T \quad (3)$$

where $L \in \mathbb{R}^{n \times n}$ is a lower triangular matrix acting as a whitening operator. To ensure numerical stability, we add a small perturbation $\epsilon I$ (where $\epsilon = 10^{-6}$) to guarantee positive definiteness. This operator $L$ projects the weights into an activation-aware manifold space.

### 3.3. Whitened Feature Decomposition

Upon obtaining the reconstruction operator $L$, we transform the original weight matrix $W$ into the whitened space as $\hat{W} = W \cdot L$. Subsequently, we perform Singular Value Decomposition (SVD) (Peng et al., 2012) on the transformed matrix $\hat{W}$:

$$\hat{W} = U\Sigma V^T \quad (4)$$

where $U \in \mathbb{R}^{m \times k}$ is the left singular vector matrix representing the feature basis of the output space, $V^T \in \mathbb{R}^{k \times n}$ is the right singular vector matrix representing the projection basis of the input space, and $\Sigma = \text{diag}(\sigma_1, \ldots, \sigma_k)$ is a diagonal matrix containing singular values in descending order with $k = \min(m, n)$. In this whitened space, the magnitude of the singular values $\sigma_i$ reflects the effective feature intensity integrated with activation energy.

### 3.4. Non-uniform Rank Allocation

Building upon the whitened feature decomposition, we translate the derived singular value spectra into a concrete rank configuration. This allocation process is governed by three pivotal questions concerning layer importance, module sensitivity, and budget alignment, which are addressed in the subsequent subsections.

#### 3.4.1. LAYER IMPORTANCE QUANTIFICATION

> **Question 1:** *Do all layers in Large Language Models exhibit equal significance? What metrics should be used for assessment?*

Recent studies (Zhang et al., 2024; Yin et al., 2024; Chen et al., 2025c) show that the importance of layers in LLMs is highly non-uniform. Shallow layers play a critical role in storing knowledge and retrieving information, whereas deeper layers often exhibit high representational redundancy, with features changing slowly across layer indices (Men et al., 2025; Gromov et al., 2025). Among the evaluation metrics, cosine similarity is widely utilized to assess the similarity between layer-wise inputs and outputs (Jiang et al.,

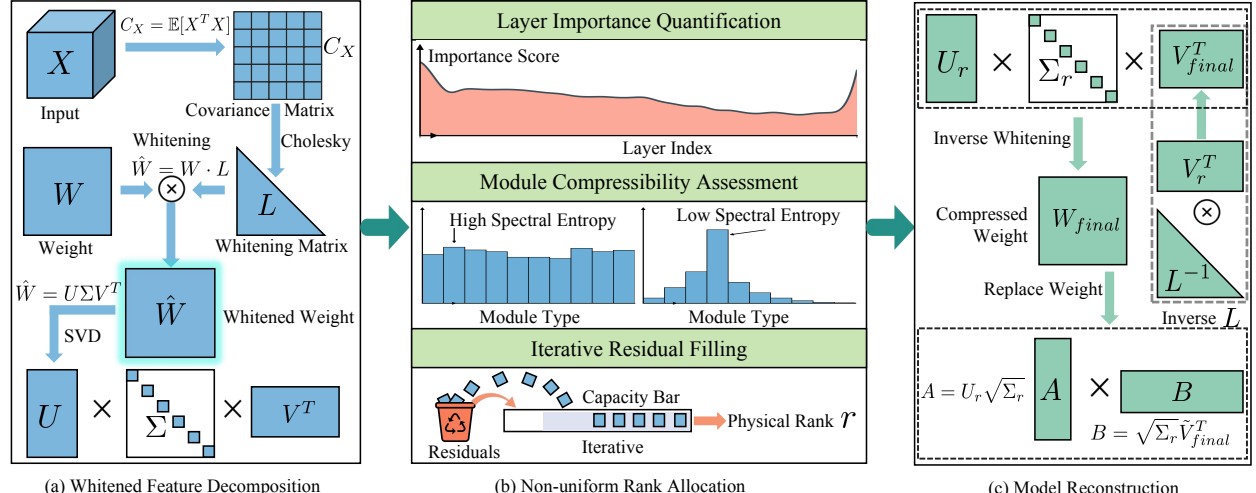

(a) Whitened Feature Decomposition  (b) Non-uniform Rank Allocation  (c) Model Reconstruction

*Figure 1.* **The schematic overview of the proposed CGSVD framework.** The compression pipeline proceeds in three distinct stages: **(a) Whitened Feature Decomposition:** Activation statistics from input $X$ are leveraged to compute the covariance matrix $C_X$ and the whitening operator $L$. The weight matrix $W$ is then projected into an activation-aware manifold ($\hat{W}$) prior to SVD to align singular values with feature importance. **(b) Non-uniform Rank Allocation:** A layer-wise decision mechanism optimizes the physical rank $r$. It sequentially quantifies inter-layer sensitivity via importance scores (top), assesses intra-layer compressibility using spectral entropy (middle), and ensures precise budget alignment through iterative residual filling (bottom). **(c) Model Reconstruction:** The truncated singular components are restored via inverse whitening matrix ($L^{-1}$) and reorganized into two cascaded low-rank linear modules, $A$ and $B$, for efficient inference.

2024; Chen et al., 2025b). While cosine similarity suffers from vanishing sensitivity near perfect alignment, angular distance maintains linear sensitivity to subtle representational shifts. To quantify this heterogeneity, we adopt the angular distance $d$ as the core metric to measure the evolution of representations between layers. Specifically, we define the importance score of the $l$-th layer, denoted as $S_l$, as the angular distance between its input and output representations, $S_l = d(x^{(l)}, x^{(l+1)})$. For a sequence of length $T$, the angular distance $d$ between layer $l$ and $l+1$ is calculated as:

$$S_l = d(x^{(l)}, x^{(l+1)}) = \frac{1}{\pi} \arccos\left(\frac{x_T^{(l)} \cdot x_T^{(l+1)}}{\|x_T^{(l)}\| \|x_T^{(l+1)}\|}\right) \quad (5)$$

where $x_T^{(l)}$ denotes the multi-dimensional representation vector of the last token $T$ at the input of layer $l$, $\|\cdot\|$ denotes the $L^2$ norm of a vector. The use of the last token is justified by the causal attention mask, which ensures its embedding depends on the entire sequence. A small $d(x^{(l)}, x^{(l+1)})$ indicates slow representational evolution and lower importance $S_l$, allowing for a reduced parameter budget in those layers.

### 3.4.2. MODULE COMPRESSIBILITY ASSESSMENT

> **Question 2:** *Do sub-modules within a layer exhibit identical compressibility? How can we quantify their low-rank approximation potential?*

Even within a single Transformer layer, different projection modules, such as the attention mechanism's $q\_proj$

and the MLP block's $down\_proj$, play distinct roles and exhibit varying singular value energy distributions (Yuan et al., 2023). To quantify this module-level functional difference, we introduce spectral entropy $H$ to assess the low-rank approximation potential. We first map the $j$-th squared singular values $\sigma_j$ to a normalized energy probability distribution $p_j$:

$$p_j = \sigma_j^2 / \sum_{i=1}^{k} \sigma_i^2 \quad (6)$$

The module-level spectral entropy $H$ is then defined according to the Shannon entropy (Lin, 1991) formulation:

$$H = -\sum_{j=1}^{k} p_j \ln(p_j) \quad (7)$$

The spectral entropy $H$ measures the dispersion of the energy distribution. A high $H$ value suggests a flat singular value distribution where information is dispersed across many dimensions, indicating that the module is difficult to approximate at a very low rank.

### 3.4.3. MULTI-LEVEL ALLOCATION AND ITERATIVE RESIDUAL FILLING

> **Question 3:** *Given a fixed global parameter budget, how should the retention ratio be allocated to each module in each layer?*

To maximize precision under a specific hardware constraint, we design a cascaded allocation decision chain. Note that

while users typically specify a compression ratio $\gamma$ (e.g., removing 20% of parameters), our internal allocation logic operates on the target retention ratio $\rho_{target} = 1 - \gamma$ (e.g., keeping 80%). This ensures that layers with higher importance scores are allocated a higher retention proportion.

In the first level, inter-layer budget adjustment is performed by dynamically tuning the target retention ratio of each layer based on its importance score $S_l$. Given a layer fluctuation ratio $\lambda_{layer}$, the target retention ratio $\rho_l$ for layer $l$ is calculated as:

$$\rho_l = \text{clip}(\rho_{target} + \lambda_{layer} \cdot (S_l - \bar{S}), \rho_{min}, \rho_{max}) \quad (8)$$

where $\bar{S}$ is the mean layer importance across the model. A positive deviation $(S_l - \bar{S} > 0)$ increases the retention budget for critical layers. $\rho_{min}$ and $\rho_{max}$ denote the lower and upper bounds of the retention ratio, respectively, ensuring that even redundant layers retain minimum functional capacity while preventing memory overflow in significant layers. $\text{clip}(\cdot, \rho_{min}, \rho_{max})$ denotes a clamping function, which mathematically serves as a projection operator that maps the variable onto the predefined closed interval $[\rho_{min}, \rho_{max}]$.

In the second level, we determine the intra-layer rank by further distributing the layer budget $\rho_l$ among modules using their spectral entropy $H_{l,m}$ and a module fluctuation ratio $\lambda_{module}$. The target retention ratio $\rho_{l,m}$ and the corresponding initial physical rank $r_{l,m}$ for module $m$ are derived as:

$$\rho_{l,m} = \text{clip}(\rho_l + \lambda_{module} \cdot (H_{l,m} - \bar{H}_l), \rho_{min}, \rho_{max}) \quad (9)$$

$$r_{l,m} = \left\lfloor \frac{\rho_{l,m} \cdot (m_{rows} \cdot n_{cols})}{m_{rows} + n_{cols}} \right\rfloor \quad (10)$$

where $\bar{H}_l$ is the mean spectral entropy within layer $l$, and $m_{rows}, n_{cols}$ are the dimensions of the weight matrix. The floor function $\lfloor \cdot \rfloor$ is necessary to obtain an integer rank, but it inevitably leads to a parameter gap where the actual retained parameter count $P_{actual}$ is less than the target budget $P_{target}$.

To fully utilize this remaining capacity, we execute an Iterative Residual Filling (IRF) mechanism. We quantify the parameter gap $\Delta P = P_{target} - P_{actual}$ and sort unsaturated modules by a filling weight $W_{fill}$:

$$W_{fill} = (m_{rows} \cdot n_{cols}) \times S_l \times H_{l,m} \quad (11)$$

This weight ensures that surplus budget is prioritized for modules that are large, layer-significant, and structurally complex. The rank increase $\Delta r$ is distributed proportionally:

$$\Delta r = \min \left( \left\lfloor \frac{\Delta P \cdot (W_{fill} / \sum W_{fill})}{m_{rows} + n_{cols}} \right\rfloor, r_{max} - r_{curr} \right) \quad (12)$$

where $\sum W_{fill}$ represents the cumulative weight of unsaturated modules, $r_{max}$ denotes the upper bound of the physical rank (constrained by the no-inflation threshold), and $r_{curr}$ denotes the rank assigned prior to the current filling step. This process repeats iteratively until the final retention ratio strictly converges to $\rho_{target}$.

## 3.5. Model Reconstruction

After determining the final physical rank $r$, we execute weight truncation and structural reconstruction. We first utilize the inverse whitening matrix $L^{-1}$ to eliminate the spatial distortion introduced in Section 3.2, yielding the recovered basis matrix $\tilde{V}_{final}^T = V_r^T \cdot L^{-1}$, where $V_r^T$ contains the top $r$ rows of the right singular vectors. We then perform low-rank reorganization by distributing the square root of the truncated singular value matrix $\Sigma_r$ equally to form two consecutive linear layers:

$$A = U_r \sqrt{\Sigma_r} \in \mathbb{R}^{m \times r} \quad (13)$$

$$B = \sqrt{\Sigma_r} \tilde{V}_{final}^T \in \mathbb{R}^{r \times n} \quad (14)$$

Finally, the original dense layer $W$ is replaced by the cascaded structure $A \times B$. This structured reorganization effectively achieves a leap in optimization, reducing computational complexity from $O(mn)$ to $O(r(m + n))$ during the inference phase. The complete execution pipeline of the proposed method is formally summarized in Appendix B.

## 4. Experiments

### 4.1. Experimental Setup

#### 4.1.1. MODELS AND DATASETS

We evaluate the performance of CGSVD on various LLMs, including LLaMA1-7B (Touvron et al., 2023a), LLaMA2 (7B/13B) (Touvron et al., 2023b), LLaMA3.1-8B (Grattafiori et al., 2024), LLaMA3.2-3B, and Vicuna-7B (Peng et al., 2023). Additionally, we evaluate the language modeling capabilities and zero-shot performance of compressed LLMs. Specifically, we measure language modeling performance using the perplexity metric on the WikiText2 (Merity et al., 2017) and C4 (Raffel et al., 2020) validation datasets. For zero-shot evaluation, we assess accuracy on six commonsense benchmarks from EleutherAI LM Harness (Gao et al., 2024a), including HellaS (Zellers et al., 2019), WinoG (Sakaguchi et al., 2021), ARC-e and ARC-c (Boratko et al., 2018), PIQA (Bisk et al., 2020), and OBQA (Mihaylov et al., 2018).

#### 4.1.2. BASELINES

To evaluate the effectiveness of CGSVD, we benchmark it against Vanilla SVD (Sainath et al., 2013) and leading SVD-based LLM compression techniques: FWSVD (Hsu et al.,

2022), ASVD (Yuan et al., 2023), SVD-LLM (Wang et al., 2025c) (Section 4.2), AdaSVD (Li et al., 2025), ResSVD (Bai et al., 2025), and Basis Sharing (Wang et al., 2025a) (Appendix D).

### 4.1.3. Implementation Details

To ensure a fair comparison, we follow the protocols of ASVD and SVD-LLM, randomly selecting 256 calibration samples with a sequence length of 2048 from WikiText2. In our implementation, we set the lower and upper bounds of the retention ratio to $\rho_{\min} = 0.05$ and $\rho_{\max} = 0.95$, respectively. These two bounds are shared by both the layer-wise and module-wise allocation stages to prevent extremely small or excessively large retention ratios, thereby maintaining stable rank allocation across different compression settings. All experiments are conducted on two NVIDIA A40 48GB GPUs.

### 4.2. Main Results

We evaluate the overall performance of the CGSVD method from four perspectives: (1) different compression rates, (2) generalizability across diverse LLM architectures, (3) practical inference efficiency, and (4) recoverability via parameter-efficient fine-tuning.

**Performance under Different Ratios.** We compare the performance of CGSVD against baseline methods on LLaMA2-7B under compression ratios ranging from 30% to 60% across eight datasets. As presented in Table 1, CGSVD consistently demonstrates superior performance compared to all baseline methods across the entire range of compression ratios. In terms of language modeling capabilities, our method maintains significantly lower perplexity on WikiText2 and C4. Notably, while standard decomposition methods (SVD, FWSVD, ASVD) suffer from catastrophic performance degradation, evidenced by perplexity scores frequently exploding into the thousands, CGSVD effectively preserves the model's linguistic coherence. For instance, under a 60% compression ratio, our method maintains a perplexity of 49.92 on WikiText2, significantly outperforming the strongest baseline, SVD-LLM, which degrades to 89.95. Furthermore, on the six zero-shot classification benchmarks, CGSVD attains the highest average accuracy at every compression level. At the 30% ratio, our method secures an average accuracy of 43.43%, surpassing SVD-LLM (39.73%) by a substantial margin of 3.70%. Even as the compression ratio increases to 60%, where all models experience a decline in capability, CGSVD remains the most robust. These results validate that the cascaded granular allocation strategy enables more precise identification and preservation of critical structures during the compression process.

**Performance on Different LLMs.** To demonstrate the scalability and generalizability of CGSVD, we evaluate our framework across five representative models spanning different architectures and parameter scales, ranging from 3B to 13B. This includes the LLaMA1-7B, Vicuna-7B, LLaMA2-13B, and the recent LLaMA3.1-8B and LLaMA3.2-3B models. As shown in Table 2, CGSVD consistently maintains the lowest perplexity and highest average zero-shot accuracy across all tested architectures at a 30% compression ratio. Notably, while traditional methods like FWSVD encounter Out-of-Memory (OOM) errors on larger models such as LLaMA2-13B, and Vanilla SVD exhibits catastrophic failure (with perplexity exceeding 10,000), CGSVD remains robust. Most significantly, CGSVD demonstrates substantial superiority over the state-of-the-art SVD-LLM on the highly optimized LLaMA3 series. On LLaMA3.1-8B, our method reduces perplexity by 15.46 and improves average zero-shot accuracy by 6.08%. Similarly, on the LLaMA3.2-3B model, CGSVD achieves a remarkable 66.94 reduction in perplexity and a 3.76% boost in average performance compared to SVD-LLM. These results confirm that our cascaded granular allocation strategy is particularly effective at preserving the capabilities of modern, high-density LLMs where uniform allocation strategies struggle. For more detailed information, please refer to Appendix H.

**Inference Speedup.** To evaluate the practical deployment efficiency of CGSVD, we compare the generation throughput of the compressed LLaMA2-7B model with SVD-LLM on a single NVIDIA A40 GPU. As shown in Figure 2, CGSVD consistently delivers higher throughput than SVD-LLM across all compression ratios. As the compression ratio increases from 30% to 60%, both methods exhibit improved throughput, while CGSVD maintains a clear advantage throughout the entire range. Specifically, CGSVD improves throughput by 28.03%–30.57% over SVD-LLM, indicating that its cascaded rank-allocation strategy more effectively converts parameter reduction into practical inference acceleration. These results demonstrate that CGSVD not only reduces model redundancy but also yields tangible deployment benefits by improving inference efficiency on GPU hardware.

**Performance with LoRA Fine-Tuning.** To further validate the recoverability of our compressed models, we evaluate the performance following LoRA fine-tuning (Hu et al., 2022). As illustrated in Figure 3, while LoRA universally alleviates the degradation caused by compression, CGSVD consistently maintains a performance advantage over the SVD-LLM baseline across all ratios. Notably, under the aggressive 60% compression scenario, our method combined with LoRA achieves a perplexity of 16.72, surpassing the fine-tuned SVD-LLM (18.53) and demonstrating a significant recovery from the pre-finetuning state (49.92). Fur-

*Table 1.* Performance of LLaMA2-7B with CGSVD (Ours) and baselines under compression ratios ranging from 30% to 60%. All methods are evaluated without fine-tuning. AVG(%) denotes the average performance of the model on the commonsense reasoning datasets. The best performance result is indicated in bold.

| RATIO | METHOD | WIKITEXT2↓ | C4↓ | OBQA | ARC-E | WINOG | HELLAS | ARC-C | PIQA | AVG↑ |
|---|---|---|---|---|---|---|---|---|---|---|
| | SVD | 19183.64 | 22953.38 | 11.40 | 25.67 | 49.80 | 25.79 | 21.16 | 51.03 | 30.81 |
| | FWSVD | 77.95 | 98.68 | 16.80 | 34.09 | 53.75 | 28.81 | 22.53 | 57.18 | 35.53 |
| 30% | ASVD | 1440.31 | 1314.52 | 14.80 | 25.17 | 50.75 | 26.47 | 21.16 | 52.94 | 31.88 |
| | SVD-LLM | 10.66 | 34.97 | 21.20 | 42.80 | 56.20 | 34.52 | 22.95 | 60.72 | 39.73 |
| | OURS | **9.37** | **23.09** | 22.00 | 51.73 | 58.48 | 37.88 | 26.79 | 63.71 | **43.43** |
| | SVD | 33346.34 | 38222.07 | 16.60 | 24.96 | 49.49 | 25.74 | 20.39 | 51.20 | 31.40 |
| | FWSVD | 720.13 | 1245.98 | 14.40 | 26.81 | 49.49 | 26.39 | 21.33 | 53.10 | 31.92 |
| 40% | ASVD | 7448.64 | 11375.65 | 13.40 | 26.3 | 49.01 | 26.09 | 21.76 | 52.18 | 31.46 |
| | SVD-LLM | 16.14 | 71.53 | 17.80 | 36.03 | 55.33 | 30.43 | 21.33 | 57.29 | 36.37 |
| | OURS | **12.74** | **43.25** | 18.60 | 39.94 | 58.33 | 33.39 | 22.53 | 60.61 | **38.90** |
| | SVD | 30430.34 | 27457.77 | 15.40 | 26.47 | 50.04 | 25.5 | 21.5 | 52.18 | 31.85 |
| | FWSVD | 5302.99 | 7744.83 | 12.60 | 26.26 | 51.22 | 26.10 | 22.01 | 52.88 | 31.85 |
| 50% | ASVD | 8603.12 | 9785.9 | 13.80 | 26.35 | 47.67 | 25.82 | 21.16 | 52.18 | 31.16 |
| | SVD-LLM | 33.29 | 197.19 | 13.80 | 31.40 | 53.51 | 28.11 | 19.62 | 55.33 | 33.63 |
| | OURS | **21.92** | **104.44** | 15.40 | 33.21 | 56.43 | 29.51 | 20.99 | 56.80 | **35.39** |
| | SVD | 48651.71 | 68329 | 16.60 | 25.84 | 52.33 | 25.28 | 22.27 | 51.85 | 32.36 |
| | FWSVD | 4878.65 | 7484.24 | 16.00 | 25.88 | 51.30 | 25.64 | 23.46 | 52.94 | 32.54 |
| 60% | ASVD | 26116.47 | 24079.13 | 12.40 | 24.49 | 49.64 | 25.67 | 22.35 | 52.99 | 31.26 |
| | SVD-LLM | 89.95 | 563.42 | 12.80 | 27.06 | 49.80 | 26.79 | 20.39 | 53.48 | 31.72 |
| | OURS | **49.92** | **267.06** | 14.20 | 29.21 | 54.46 | 27.43 | 20.05 | 55.28 | **33.44** |

*Table 2.* Performance comparison of different LLMs under a 30% compression ratio. PPL denotes the perplexity evaluated on the WikiText2 dataset, and AVG(%) represents the average performance on commonsense reasoning datasets. All methods are evaluated without fine-tuning. OOM indicates an out-of-memory error.

| METHOD | LLAMA1-7B | | VICUNA-7B | | LLAMA2-13B | | LLAMA3.1-8B | | LLAMA3.2-3B | |
|---|---|---|---|---|---|---|---|---|---|---|
| | PPL↓ | AVG↑ | PPL↓ | AVG↑ | PPL↓ | AVG↑ | PPL↓ | AVG↑ | PPL↓ | AVG↑ |
| SVD | 9232.6 | 32.13 | 2383.31 | 31.83 | 18287.59 | 31.10 | 323676.55 | 31.84 | 161488.22 | 31.70 |
| FWSVD | 52.67 | 37.12 | 46.12 | 39.38 | OOM | OOM | 3709.86 | 31.43 | 1092.36 | 31.33 |
| ASVD | 243.16 | 33.61 | 132.68 | 33.60 | 26.74 | 40.79 | 20530.92 | 31.06 | 2811.31 | 32.24 |
| SVD-LLM | 9.52 | 43.15 | 12.42 | 42.51 | 8.00 | 45.82 | 32.92 | 36.41 | 95.08 | 33.74 |
| OURS | **8.79** | **46.33** | **11.21** | **44.93** | **7.49** | **47.99** | **17.46** | **42.49** | **28.14** | **37.50** |

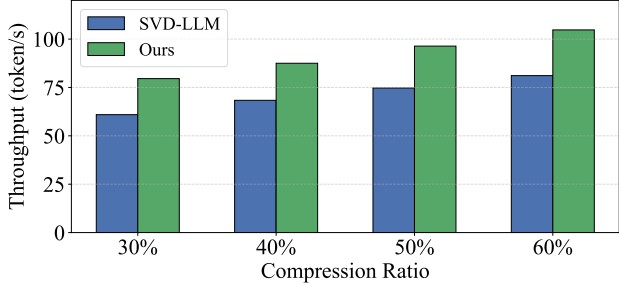

*Figure 2.* Throughput of Compressed LLaMA2-7B on a single NVIDIA A40 GPU.(Batch Size=4, Seq Length=128).

thermore, even at 30% ratio, CGSVD with LoRA reaches a perplexity of 8.23, outperforming the baseline's 8.92. These results confirm that the semantic structure preserved by our granular allocation strategy provides a superior initialization point, enabling parameter-efficient fine-tuning to more effectively recover model capabilities.

### 4.3. Ablation Study

In this section, we conduct a systematic ablation analysis to evaluate the individual effectiveness of the three core components integrated into our CGSVD framework: the **Layer Importance (LI)** quantification via angular distance, the **Module Compressibility (MC)** assessment via spectral entropy, and the **Iterative Residual Filling (IRF)** mechanism for budget alignment.

We evaluate the perplexity of the LLaMA2-7B model on the WikiText2 dataset across four target compression ratios $\gamma \in \{30\%, 40\%, 50\%, 60\%\}$. The results are summarized in Table 3. In the table, "SVD-based" denotes the baseline uniform rank allocation strategy as implemented in SVD-LLM. "+LI" indicates the application of the inter-layer budget adjustment (Eq. 8) based on the angular distance $S_l$. "+MC" denotes the exclusive application of the intra-layer rank distribution strategy (Eq. 9), where the retention budget is allocated to specific modules based on their spectral

*Table 3.* Ablation study of CGSVD components on LLaMA2-7B. Performance is measured by perplexity (PPL) on the WikiText2 dataset. LI: Layer Importance; MC: Module Compressibility; IRF: Iterative Residual Filling.

| METHOD | 30% | 40% | 50% | 60% |
|---|---|---|---|---|
| SVD-BASED | 10.66 | 16.14 | 33.29 | 89.95 |
| + LI | 10.37 | 14.78 | 26.82 | 65.85 |
| + MC | 9.95 | 14.51 | 28.09 | 73.83 |
| + LI + MC | 9.56 | 12.94 | 22.09 | 50.48 |
| + LI + MC + IRF | **9.37** | **12.74** | **21.92** | **49.92** |

entropy $H_{l,m}$ to prioritize components with higher information density, while maintaining a uniform target ratio across layers. "+LI+MC" further incorporates the intra-layer rank distribution utilizing the module-level spectral entropy $H_{l,m}$ to assess the low-rank approximation potential. Finally, "+LI+MC+IRF" represents our complete CGSVD pipeline, which combines the dual-level non-uniform allocation with the iterative residual filling mechanism (Eq. 11–12) to bridge the parameter gap $\Delta P$ and ensure the final model size strictly adheres to the global target budget defined by $\gamma$.

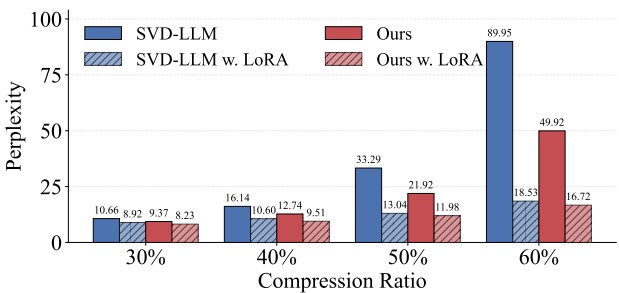

*Figure 3.* Perplexity comparison on the WikiText2 dataset across varying compression ratios (30%-60%).

As shown in Table 3, all three strategies contribute significantly to the reduction of perplexity, and their performance gains become increasingly pronounced at higher compression ratios. Specifically, while the LI strategy effectively captures the macroscopic redundancy across different depths, the MC provides the necessary fine granularity to preserve information in structurally complex modules with high spectral entropy. The IRF mechanism further ensures that the actual parameter count $P_{actual}$ fully utilizes the available capacity by prioritizing modules with high filling weights $W_{fill}$. These findings validate the cascaded design of CGSVD, demonstrating that the synergy of layer-wise allocation and precise residual filling is essential for maintaining model performance.

In addition to the component-wise ablation, we further analyze the sensitivity of CGSVD to calibration data configuration. As shown in Figure 4, increasing the calibration batch size consistently improves the compressed model per-

formance on both WikiText2 and C4, with perplexity decreasing from 11.31 to 9.29 on WikiText2 and from 19.22 to 14.93 on C4. This indicates that a larger number of calibration samples provides more reliable activation statistics for whitening-based reconstruction and rank allocation. We observe a similar but more pronounced trend when increasing the calibration sequence length, where perplexity is reduced from 20.28 to 8.65 on WikiText2 and from 34.67 to 14.25 on C4, suggesting that longer sequences better capture contextual dependencies and improve the estimation of layer-wise and module-wise importance. Notably, the performance gains are substantial under small calibration settings, while the improvements gradually diminish as the batch size and sequence length become larger. These results demonstrate that CGSVD benefits from richer calibration data but remains stable under moderate calibration configurations, confirming its robustness and practicality for training-free LLM compression.

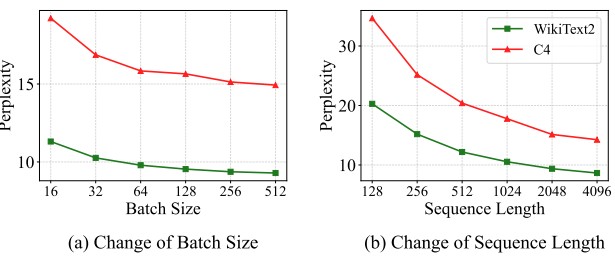

(a) Change of Batch Size     (b) Change of Sequence Length

*Figure 4.* Ablation study on calibration data configuration. (a) The impact of calibration batch size on perplexity over WikiText2 and C4 datasets. (b) The impact of calibration sequence length on perplexity over WikiText2 and C4 datasets.

### 4.4. More Corroborating Results

**Integration with Weight Quantization.** Since SVD-based methods fundamentally alter the model topology by replacing dense layers with low-rank factors, they are orthogonal to weight-quantization methods that reduce numerical precision. To systematically evaluate compatibility and the practical limits of compound compression, we integrate GPTQ (Frantar et al., 2022) with CGSVD on LLaMA2-7B. We evaluate the perplexity on WikiText2 under 16-bit, 8-bit, and 4-bit quantization settings across compression ratios from 30% to 50%.

As illustrated in Figure 5, CGSVD exhibits exceptional robustness when integrated with quantization. Remarkably, 8-bit quantization introduces negligible performance degradation compared to the 16-bit baseline. This stability indicates that the singular components preserved by CGSVD capture the essential variance of the weights, making them resilient to quantization noise. Furthermore, while 4-bit quantization introduces a visible perplexity increase (rising to 14.31 at the 30% compression ratio), the model avoids catastrophic failure, demonstrating that CGSVD can serve

as a viable structural pre-compressor.

**Comparison with recent methods & Efficiency.** To comprehensively evaluate the competitiveness of CGSVD, we provide a detailed comparison with recent state-of-the-art methods, such as AdaSVD and ResSVD, in Appendix D. The results demonstrate that our approach achieves superior performance. Furthermore, the speed evaluation in Appendix G corroborates the computational efficiency of CGSVD, highlighting that it significantly outperforms ASVD in processing speed, thereby greatly facilitating rapid deployment in practical scenarios.

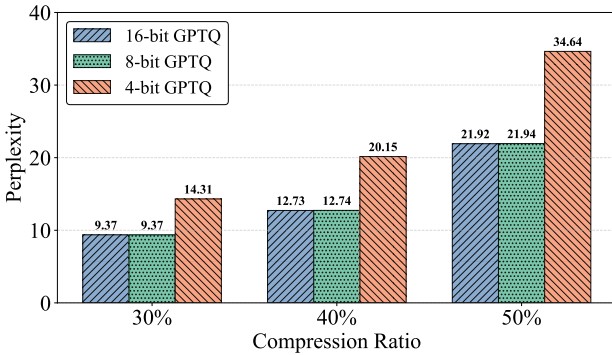

*Figure 5.* Impact of Weight Quantization on CGSVD. We evaluate the perplexity of LLaMA2-7B on WikiText2 across compression ratios of 30%-50%.

## 5. Conclusion

In this paper, we present CGSVD, a high-efficiency framework for LLM compression that leverages cascaded granular singular value decomposition to maximize semantic preservation. Specifically, CGSVD introduces a dual-level non-uniform rank allocation strategy: it utilizes angular distance to quantify inter-layer representational shifts and employs spectral entropy to assess the intrinsic compressibility of specific projection modules. This layer-wise decision-making process, coupled with an iterative residual filling mechanism, ensures precise budget alignment and optimal parameter utilization without requiring computationally expensive retraining or fine-tuning. Extensive empirical results demonstrate that CGSVD sets a new state-of-the-art for training-free low-rank decomposition, maintaining robust linguistic coherence and reasoning capabilities.

## Impact Statement

This work contributes to the democratization of Large Language Models by significantly lowering the hardware barriers required for their deployment. By enabling high-fidelity model compression without the need for computationally expensive retraining, CGSVD facilitates the proliferation of ad-

vanced AI capabilities to resource-constrained environments and edge devices. This shift not only promotes privacy-preserving local inference but also aligns with Green AI initiatives by reducing the energy footprint associated with large-scale model serving. However, we acknowledge that increasing the accessibility of powerful generative models simultaneously lowers the threshold for potential misuse, such as the automated generation of disinformation or spam by actors with limited computational resources. Furthermore, while our method preserves semantic consistency, the subtle effects of non-uniform rank truncation on safety guardrails and latent biases warrant continuous vigilance. We encourage the research community to rigorously evaluate the safety alignment of compressed architectures alongside their computational efficiency.

## Acknowledgments

We would like to thank the anonymous reviewers, our program chairs, senior area chairs, and area chairs for their thoughtful comments and support on this work. This work was supported by the National Natural Science Foundation of China (Grant No.62372058, U22A2026).

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

## A. Automatic Fluctuation Ratio Search via Perplexity Optimization

While the layer-wise allocation strategy effectively distributes the parameter budget, the sensitivity of model performance to the fluctuation ratios $\lambda_{layer}$ and $\lambda_{module}$ in Eq. 8 and 9 can vary significantly across different model architectures and target compression ratios $\rho_{target}$. To eliminate the heuristic manual tuning of these hyperparameters, we propose an automatic search mechanism based on calibration perplexity (PPL).

We define the search for optimal fluctuation ratios as a constrained optimization problem:

$$(\lambda_{layer}^*, \lambda_{module}^*) = \arg \min_{\lambda_L, \lambda_M} \text{PPL}(\mathcal{M}'(\lambda_L, \lambda_M); \mathcal{C}_{eval}) \tag{15}$$

where $\mathcal{M}'(\lambda_L, \lambda_M)$ represents the model compressed under the configuration $(\lambda_L, \lambda_M)$, and $\mathcal{C}_{eval}$ is a subset of the calibration dataset.

To maintain computational efficiency, we employ a Coordinate Descent search strategy rather than an exhaustive grid search. The process is divided into two sequential phases:

- **Phase 1: Layer-wise Sensitivity Search.** We fix the module fluctuation ratio $\lambda_{module}$ at its median value and iterate through a candidate set $\Lambda_L \in [0.1, 0.9]$. For each $\lambda_{layer} \in \Lambda_L$, we perform a trial rank allocation and reconstruct the weights to evaluate the resulting PPL.

- **Phase 2: Intra-layer Granularity Search.** After identifying the optimal $\lambda_{layer}^*$, we fix it and search for $\lambda_{module}$ within the candidate set $\Lambda_M$.

To further accelerate this process, we implement Inflection Point Detection (Christopoulos, 2016). Since the relationship between the fluctuation ratio and PPL typically exhibits a convex trend, the search for a specific dimension is prematurely terminated if the observed PPL begins to increase, indicating that the local optimum has been surpassed. This search mechanism ensures that CGSVD adaptively finds the optimal balance between global consistency and local heterogeneity for any given LLM.

## B. Pseudocode of CGSVD

The compression process of CGSVD is formally defined in Algorithm 1. The framework operates through a sequential pipeline consisting of four primary phases: first, it performs spatial reconstruction by projecting weights into an activation-aware whitened manifold; second, it conducts dual-level sensitivity analysis by quantifying layer-wise importance through angular distance and assessing module compressibility via spectral entropy; third, it executes hierarchical rank allocation alongside iterative residual filling to satisfy the global parameter budget precisely; and finally, it achieves model reconstruction through inverse whitening and low-rank reorganization.

## C. Layer Importance and Spectral Entropy Analysis

To motivate the necessity of non-uniform and fine-grained rank allocation in CGSVD, we conduct an empirical analysis of layer-wise importance and module-level spectral entropy on LLaMA2-7B. Figure 6 visualizes two complementary perspectives: (a) the distribution of representation evolution across network depth, and (b) the spectral entropy of different projection modules within selected layers.

**Layer-wise Importance Distribution.** Figure 6(a) presents the layer importance scores $S_l$ measured by the angular distance between consecutive layer representations. The results reveal a pronounced non-uniform pattern along the depth of the model. Early layers (e.g., Layers 0 and 1) exhibit significantly higher importance scores, indicating rapid representational transformation and strong semantic grounding. In contrast, middle layers show a relatively flat and lower importance profile, suggesting slower representation evolution and higher redundancy. Interestingly, the final layers (Layers 30 and 31) again demonstrate elevated importance, reflecting their critical role in aggregating global context and shaping the final output distribution.

This U-shaped importance profile corroborates recent findings that Transformer representations evolve non-monotonically with depth, and directly challenges the assumption of uniform compressibility across layers. Such heterogeneity implies that

a global, uniform rank truncation strategy is inherently suboptimal, as it risks over-compressing semantically critical layers while under-utilizing redundancy in less sensitive regions.

**Module-level Spectral Entropy Characteristics.** While layer importance captures macroscopic redundancy, Figure 6(b) further exposes substantial intra-layer heterogeneity across different projection modules. The spectral entropy $H$ varies not only across layers but also across module types within the same layer. In particular, self-attention projection matrices (e.g., q_proj, k_proj, and v_proj) generally exhibit higher spectral entropy than MLP projections in earlier layers, indicating a more dispersed singular value spectrum and reduced low-rank approximation potential.

Conversely, MLP modules, especially down_proj in deeper layers, often show lower spectral entropy, suggesting that their information content is concentrated in fewer dominant singular directions. This observation implies that even within a layer deemed important, certain modules can still tolerate aggressive rank reduction without severe information loss. Notably, the entropy patterns are not strictly depth-aligned; some mid-layer attention modules exhibit higher entropy than their shallow or deep counterparts, highlighting the insufficiency of depth-based heuristics alone.

**Implications for Cascaded Rank Allocation.** The joint analysis of Figures 6(a) and (b) reveals a two-level structural imbalance in LLMs: inter-layer semantic importance varies significantly across depth, while intra-layer modules exhibit distinct low-rank characteristics. These findings directly motivate the cascaded allocation strategy adopted in CGSVD. Layer-wise importance scores $S_l$ guide the coarse-grained redistribution of the global parameter budget, ensuring that semantically critical depths are adequately preserved. Meanwhile, module-level spectral entropy $H_{l,m}$ enables fine-grained differentiation within each layer, allowing structurally complex modules to retain sufficient rank while aggressively compressing low-entropy components.

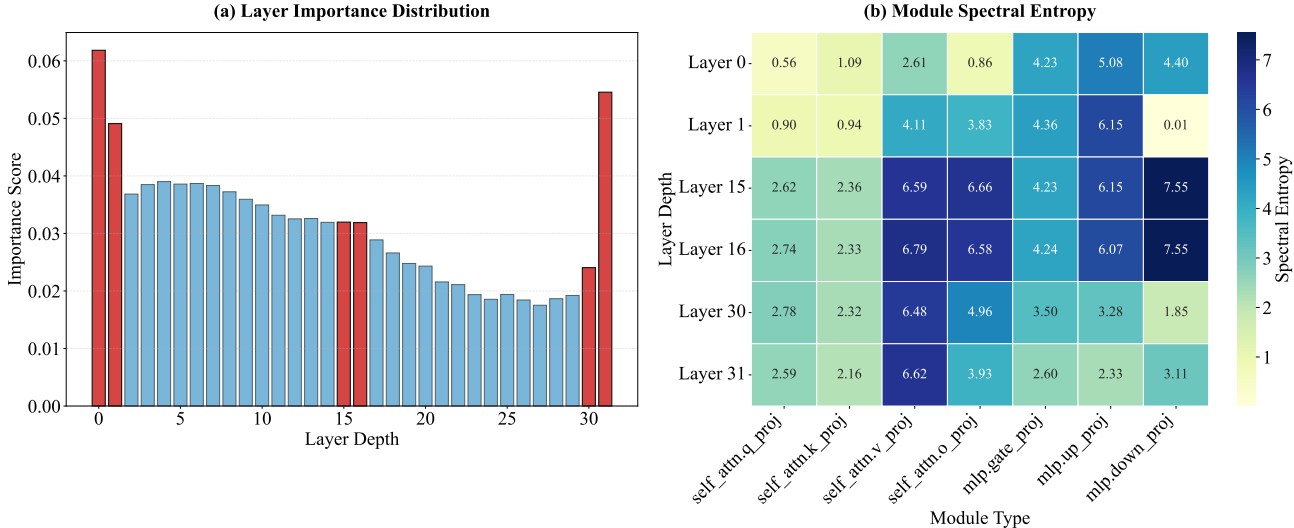

*Figure 6.* Layer importance and spectral entropy analysis for LLaMA2-7B. (a) Distribution of layer importance scores across the network depth. The red bars indicate the specific layers (Layers 0, 1, 15, 16, 30, and 31) selected for detailed visualization in the right panel. (b) Heatmap displaying the spectral entropy of different projection modules for the selected layers shown in (a).

## D. Comparison of AdaSVD and ResSVD

To further substantiate the superiority of CGSVD, we extend our evaluation to include recent state-of-the-art SVD-based compression methods, including AdaSVD (Li et al., 2025), ResSVD (Bai et al., 2025), and Basis Sharing (Wang et al., 2025a). Since the official implementations of AdaSVD and ResSVD are not publicly available, we follow the experimental settings described in their original papers and reference their reported perplexity results on LLaMA2-7B using the WikiText2 dataset. For Basis Sharing, we reproduce the results based on its open-source implementation under the same evaluation setting. We focus on the high-compression regime with compression ratios of 40%, 50%, and 60%, as summarized in

*Table 4.* Comparative analysis of perplexity on the WikiText2 dataset for LLaMA2-7B. CGSVD outperforms recent state-of-the-art SVD-based compression methods, including AdaSVD, ResSVD, and Basis Sharing, across all compression ratios, demonstrating the effectiveness of the proposed granular rank-allocation strategy.

| METHOD | 40% | 50% | 60% |
|---|---|---|---|
| SVD-LLM (WANG ET AL., 2025C) | 16.14 | 33.29 | 89.95 |
| ADASVD (LI ET AL., 2025) | 14.76 | 25.58 | 50.33 |
| RESSVD (BAI ET AL., 2025) | 14.17 | 24.26 | 58.88 |
| BASIS SHARING (WANG ET AL., 2025A) | 13.87 | 23.77 | 50.38 |
| OURS | **12.74** | **21.92** | **49.92** |

*Table 5.* Comparison of different layer-importance criteria on LLaMA2-7B under a 30% compression ratio. Performance is measured by perplexity on the WikiText2 dataset.

| METHOD | ANGULAR-AVG | NORM CHANGE | RECON. ERROR | COSINE | OURS |
|---|---|---|---|---|---|
| PPL | 9.41 | 9.99 | 10.32 | 9.40 | **9.37** |

Table 4.

The results show that CGSVD consistently achieves the lowest perplexity across all compression ratios. Compared with SVD-LLM (Wang et al., 2025c), AdaSVD, ResSVD, and Basis Sharing, CGSVD maintains a clear advantage in both moderate and aggressive compression settings. In particular, although Basis Sharing already improves upon SVD-LLM by reusing low-rank bases, CGSVD still further reduces perplexity from 13.87 to 12.74 at 40% compression and from 23.77 to 21.92 at 50% compression. At 60% compression ratio, CGSVD also obtains the best perplexity, slightly outperforming AdaSVD and Basis Sharing while substantially surpassing ResSVD and SVD-LLM. These results demonstrate that the proposed cascaded granular rank-allocation strategy can more effectively preserve critical model components under constrained parameter budgets. More importantly, the comparison suggests that a scientifically designed rank distribution is crucial for recovering model capability in training-free SVD-based LLM compression.

# E. Additional Analysis of Allocation Criteria

## E.1. Robustness of the Layer-Importance Signal

To further validate the robustness of the proposed layer-importance signal, we compare the angular-distance criterion used in CGSVD with several alternative indicators, including token-averaged angular distance, activation norm change, reconstruction error, and cosine similarity. For a fair comparison, all variants follow the same CGSVD pipeline and only replace the layer-importance criterion used for inter-layer rank allocation. We conduct the comparison on LLaMA2-7B under a 30% compression ratio and report the perplexity on WikiText2.

As shown in Table 5, the proposed last-token angular distance achieves the lowest perplexity among all evaluated criteria. Although token-averaged angular distance and cosine similarity also provide competitive results, the last-token angular distance yields the best performance while maintaining a simple and efficient computation. This result supports our design choice: under causal language modeling, the final token representation aggregates contextual information from the preceding sequence, making it an effective proxy for measuring layer-wise representational evolution. In contrast, activation norm change and reconstruction error lead to higher perplexity, suggesting that magnitude variation or local reconstruction discrepancy alone is less effective for identifying globally important layers. These results confirm that the proposed angular-distance-based importance signal is both robust and effective for guiding non-uniform inter-layer rank allocation.

## E.2. Justification of Spectral Entropy for Compressibility Assessment

We further justify the use of spectral entropy for module-level compressibility assessment by comparing it with a variance-based alternative. In CGSVD, module compressibility is determined by how the singular-value energy is distributed across latent dimensions. A module with energy concentrated in a few dominant singular directions is easier to approximate using a low rank, whereas a module with energy dispersed across many dimensions requires a larger rank to preserve its information. Spectral entropy directly captures this global energy dispersion by operating on the normalized singular-value energy distribution. Therefore, it is naturally aligned with the objective of low-rank approximation.

*Table 6.* Comparison between the variance-based alternative and the proposed entropy-based compressibility criterion on LLaMA2-7B. Performance is measured by perplexity on the WikiText2 dataset.

| METHOD | 30% | 40% | 50% |
|---|---|---|---|
| VARIANCE | 5994.43 | 6869.11 | 9988.90 |
| OURS | **9.37** | **12.74** | **21.92** |

Table 6 reports the comparison on LLaMA2-7B under different compression ratios. The variance-based alternative leads to severe performance degradation, with perplexity increasing to thousands even at a 30% compression ratio. In contrast, the entropy-based criterion used in CGSVD maintains substantially lower perplexity across all compression ratios. This large gap indicates that variance is not suitable for assessing module compressibility in this setting, because it mainly reflects magnitude dispersion rather than the normalized allocation of spectral energy. By contrast, spectral entropy provides a more stable and informative measure of whether the information in a module is concentrated or broadly distributed. These results empirically validate the necessity of entropy-based compressibility assessment in the proposed cascaded granular rank-allocation strategy.

## F. Hyperparameters for LoRA Fine-tuning

Table 7 presents the detailed hyperparameters and target module configurations employed for the LoRA fine-tuning stage on the Alpaca dataset (Taori et al., 2023).

*Table 7.* Hyperparameters for LoRA Fine-tuning.

| Parameter | Value |
|---|---|
| Dataset | Alpaca |
| Batch Size | 128 |
| Micro Batch Size | 4 |
| Epoch | 2 |
| Learning Rate | 1e-5 |
| LoRA-Rank | 8 |
| LoRA-Alpha | 16 |
| LoRA-Dropout | 0.05 |
| Max Length | 256 |
| Validation Dataset Size | 2000 |

## G. Compression Speed Evaluation

As shown in Table 8, CGSVD demonstrates superior computational efficiency suitable for rapid deployment, achieving a speedup of approximately $58\times$ over the ASVD baseline. The prohibitive latency of ASVD (over 58 hours) is fundamentally caused by its Sensitivity-based Truncation Rank Searching (STRS) mechanism, which necessitates a binary search for optimal ranks and requires repeated forward propagation on the calibration set to evaluate the perplexity of every candidate configuration.

In contrast, CGSVD circumvents this computationally expensive search loop by deriving sensitivity directly from intrinsic statistical properties. Specifically, we utilize angular distance and spectral entropy to quantify layer-wise and module-wise importance in a one-shot manner, eliminating the need for iterative inference. Although our total execution time (60 minutes) involves a marginal overhead compared to the simplistic SVD-LLM (38 minutes), this additional time is explicitly invested in the Iterative Residual Filling (IRF) mechanism, the granular entropy calculations, and the layer-wise importance quantification. Given that these components are decisive for our superior accuracy, this slight computational cost is negligible compared to the orders-of-magnitude reduction in computational overhead of training-based methods.

*Table 8.* Comparison of compression speed (calibration and decomposition) on LLaMA2-7B at a 30% compression ratio on a single NVIDIA A40 GPU. The total time includes both the data calibration phase and the matrix decomposition phase.

| ASVD | | | SVD-LLM | | | OURS | | |
|---|---|---|---|---|---|---|---|---|
| CALIBRATION | DECOMPOSITION | TOTAL | CALIBRATION | DECOMPOSITION | TOTAL | CALIBRATION | DECOMPOSITION | TOTAL |
| 58H28MIN | 2MIN | 58.5H | 18MIN | 20MIN | 38MIN | 31MIN | 29MIN | 60MIN |

*Table 9.* Performance comparison of different LLMs at a 30% compression ratio. AVG(%) represents the average performance on commonsense reasoning datasets. All methods are evaluated without fine-tuning. OOM indicates an out-of-memory error.

| MODEL | METHOD | WIKITEXT2↓ | C4↓ | OBQA | ARC-E | WINOG | HELLAS | ARC-C | PIQA | AVG↑ |
|---|---|---|---|---|---|---|---|---|---|---|
| | SVD | 9232.6 | 9948.64 | 16.20 | 26.35 | 50.75 | 25.45 | 21.08 | 52.94 | 32.13 |
| | FWSVD | 52.67 | 67.50 | 15.80 | 40.66 | 54.38 | 30.96 | 20.82 | 60.07 | 37.12 |
| LLAMA1-7B | ASVD | 243.16 | 229.24 | 14.40 | 30.72 | 52.33 | 28.30 | 20.65 | 55.28 | 33.61 |
| | SVD-LLM | 9.52 | 26.39 | 20.40 | 49.54 | 59.43 | 37.18 | 27.13 | 65.23 | 43.15 |
| | OURS | **8.79** | **19.22** | 23.20 | 56.02 | 62.75 | 40.23 | 28.84 | 66.92 | **46.33** |
| | SVD | 2383.31 | 2233.16 | 12.80 | 26.09 | 51.30 | 26.28 | 21.33 | 53.16 | 31.83 |
| | FWSVD | 46.12 | 65.69 | 17.80 | 47.22 | 54.14 | 31.19 | 24.06 | 61.86 | 39.38 |
| VICUNA-7B | ASVD | 132.68 | 222.92 | 14.20 | 32.53 | 51.70 | 28.33 | 19.80 | 55.06 | 33.60 |
| | SVD-LLM | 12.42 | 39.56 | 22.60 | 50.46 | 56.67 | 35.37 | 26.71 | 63.22 | 42.51 |
| | OURS | **11.21** | **27.47** | 21.40 | 54.38 | 59.67 | 39.07 | 29.61 | 65.45 | **44.93** |
| | SVD | 18287.59 | 20893.2 | 11.80 | 25.76 | 49.57 | 25.79 | 19.88 | 53.81 | 31.10 |
| | FWSVD | | | | | OOM | | | | |
| LLAMA2-13B | ASVD | 26.74 | 38.73 | 19.60 | 46.76 | 57.54 | 33.58 | 23.55 | 63.71 | 40.79 |
| | SVD-LLM | 8.00 | 24.41 | 26.20 | 54.97 | 63.54 | 37.59 | 26.71 | 65.89 | 45.82 |
| | OURS | **7.49** | **18.41** | 26.00 | 60.4 | 64.17 | 40.56 | 28.67 | 68.12 | **47.99** |
| | SVD | 323676.55 | 225962.65 | 14.80 | 24.83 | 50.99 | 25.42 | 22.53 | 52.45 | 31.84 |
| | FWSVD | 3709.86 | 3004.34 | 10.00 | 28.83 | 49.41 | 26.23 | 19.80 | 54.30 | 31.43 |
| LLAMA3.1-8B | ASVD | 20530.92 | 12452.63 | 14.00 | 25.80 | 49.49 | 25.69 | 20.22 | 51.14 | 31.06 |
| | SVD-LLM | 32.92 | 324.16 | 17.00 | 38.55 | 55.09 | 30.28 | 20.22 | 57.29 | 36.41 |
| | OURS | **17.46** | **92.22** | 19.80 | 49.28 | 61.17 | 36.03 | 25.00 | 63.66 | **42.49** |
| | SVD | 161488.22 | 140303.4 | 15.20 | 24.92 | 50.51 | 25.68 | 20.73 | 53.16 | 31.70 |
| | FWSVD | 1092.36 | 1522.52 | 10.40 | 26.77 | 50.67 | 26.24 | 20.05 | 53.86 | 31.33 |
| LLAMA3.2-3B | ASVD | 2811.31 | 2001.32 | 13.80 | 25.06 | 48.12 | 26.11 | 20.65 | 51.72 | 32.24 |
| | SVD-LLM | 95.08 | 317.89 | 12.60 | 34.09 | 52.17 | 28.15 | 19.37 | 56.04 | 33.74 |
| | OURS | **28.14** | **118.10** | 15.40 | 41.41 | 56.04 | 31.63 | 21.16 | 59.36 | **37.50** |

# H. Performance on Different LLMs

Table 9 provides a comprehensive breakdown of the performance across five distinct Large Language Model families under a 30% compression ratio. The results further corroborate the robustness of CGSVD across varying architectural designs and parameter scales.

In the case of the foundational LLaMA1-7B and Vicuna-7B models, CGSVD achieves remarkable gains in reasoning capabilities. Specifically, on the LLaMA1-7B model, our method boosts the accuracy on the ARC-Easy benchmark to 56.02%, significantly outperforming the strongest baseline, SVD-LLM, which achieves 49.54%. Similarly, on the WinoGrande (WinoG) dataset, CGSVD improves accuracy by 3.32% compared to SVD-LLM.

For larger-scale models such as LLaMA2-13B, where traditional methods like FWSVD fail due to memory constraints (OOM), CGSVD not only remains efficient but also pushes the limits of compression fidelity. We achieve a perplexity of 7.49 on WikiText2, surpassing SVD-LLM's 8.00, while raising the average zero-shot accuracy to 47.99%.

Most notably, the superiority of our framework is amplified on the modern, high-density LLaMA3 family. On LLaMA3.1-8B, CGSVD reduces the perplexity from 32.92 (SVD-LLM) to 17.46, representing a 46.9% relative improvement in language modeling coherence. Furthermore, on the compact LLaMA3.2-3B model, while the baseline SVD-LLM struggles with a perplexity of 95.08, CGSVD stabilizes the model at 28.14. These results indicate that our dual-level non-uniform allocation strategy effectively mitigates the sensitivity issues inherent in recent architectures that utilize Grouped Query Attention (GQA) and expansive training tokens.

---

**Algorithm 1** Pseudocode of CGSVD

---

1: **Input:** Pre-trained LLM $\mathcal{M}$ with $\mathcal{K}$ layers; Calibration data $\mathcal{C}$; Compression Ratio $\gamma$; Target Parameter Retention Ratio $\rho_{target}$ (where $\rho_{target} = 1 - \gamma$)
2: **Parameters:** Layer fluctuation ratio $\lambda_{layer}$; Module fluctuation ratio $\lambda_{module}$
3: **Output:** Compressed LLM $\mathcal{M}'$
4: CGSVD($\mathcal{M}, \mathcal{C}, \rho_{target}$)
5: **// Phase 1: Spatial Reconstruction via Activation Statistics**
6: $X \leftarrow$ Collect activation samples from calibration dataset $\mathcal{C}$
7: **for** each module $m$ in layer $l \in [1, \mathcal{K}]$ **do**
8:    $C_X = \mathbb{E}[X^T X]$          $\triangleright$ Compute covariance matrix, Eq. 2
9:    $C_X + \epsilon I = LL^T$          $\triangleright$ Cholesky decomposition for whitening, Eq. 3
10:   $\hat{W} = W \cdot L$          $\triangleright$ Project weight matrix into whitened space
11:   $\hat{W} = U\Sigma V^T$          $\triangleright$ Perform SVD in whitened manifold, Eq. 4
12: **end for**
13: **// Phase 2: Layer Importance Quantification & Module Compression Assessment**
14: **for** $l = 1$ to $\mathcal{K}$ **do**
15:   $S_l = \frac{1}{\pi} \arccos\left( \frac{x_T^{(l)} \cdot x_T^{(l+1)}}{\|x_T^{(l)}\| \|x_T^{(l+1)}\|} \right)$     $\triangleright$ Layer-wise angular distance, Eq. 5
16:   **for** each module $m$ in layer $l$ **do**
17:     $p_j = \sigma_j^2 / \sum_{i=1}^{k} \sigma_i^2$       $\triangleright$ Normalized energy distribution, Eq. 6
18:     $H = -\sum_{j=1}^{k} p_j \ln(p_j)$       $\triangleright$ Module-level spectral entropy, Eq. 7
19:   **end for**
20: **end for**
21: **// Phase 3: Multi-level Allocation and Iterative Residual Filling**
22: *Level 1: Inter-layer Budgeting*
23: $\bar{S} \leftarrow \text{Mean}(S)$
24: **for** $l = 1$ to $\mathcal{K}$ **do**
25:   $\rho_l = \text{clip}(\rho_{target} + \lambda_{layer} \cdot (S_l - \bar{S}), \rho_{min}, \rho_{max})$     $\triangleright$ Layer retention budget, Eq. 8
26: **end for**
27: *Level 2: Intra-layer Module Allocation*
28: **for** each module $m$ in layer $l$ **do**
29:   $\bar{H}_l \leftarrow \text{Mean}(H_l)$
30:   $\rho_{l,m} = \text{clip}(\rho_l + \lambda_{module} \cdot (H_{l,m} - \bar{H}_l), \rho_{min}, \rho_{max})$     $\triangleright$ Module retention budget, Eq. 9
31:   $r_{l,m} = \lfloor \frac{\rho_{l,m} \cdot (m_{rows} \cdot n_{cols})}{m_{rows} + n_{cols}} \rfloor$     $\triangleright$ Initial rank assignment, Eq. 10
32: **end for**
33: *Level 3: Iterative Residual Filling*
34: **while** $P_{actual} < P_{target}$ **and** $iter < max\_iter$ **do**
35:   $W_{fill} = (m_{rows} \cdot n_{cols}) \times S_l \times H_{l,m}$     $\triangleright$ Weighted filling priority, Eq. 11
36:   $\Delta r = \min\left( \lfloor \frac{\Delta P \cdot (W_{fill} / \sum W_{fill})}{m_{rows} + n_{cols}} \rfloor, r_{max} - r_{curr} \right)$     $\triangleright$ Residual gap distribution, Eq. 12
37:   $r_{l,m} \leftarrow r_{l,m} + \Delta r$
38: **end while**
39: **// Phase 4: Model Reconstruction**
40: **for** each module $m$ in layer $l$ **do**
41:   $\tilde{V}_{final}^T = V_r^T \cdot L^{-1}$     $\triangleright$ Inverse whitening to eliminate spatial distortion
42:   $A = U_r \sqrt{\Sigma_r}, \quad B = \sqrt{\Sigma_r} \tilde{V}_{final}^T$     $\triangleright$ Low-rank reorganization, Eq. 13-14
43:   Replace $W$ with cascaded layers $A \times B$
44: **end for**
45: **return** $\mathcal{M}'$
46: **end procedure**

---

