# OpenReview forum: "CGSVD: Cascaded Granular Singular Value Decomposition for Large Language Model Compression"
_ICML.cc/2026/Conference — ICML 2026 regular_

### Official Review · Reviewer_oq3H · 2026-02-26

**Soundness:** 2
**Presentation:** 2
**Significance:** 2
**Originality:** 2
**Overall Recommendation:** 2
**Confidence:** 4

**Summary:**

The paper introduces CGSVD, a training-free framework designed to compress LLMs via low-rank decomposition. CGSVD employs a dual-level non-uniform rank allocation approach. First, it quantifies inter-layer importance using the angular distance of representations to adjust parameter budgets across the model's depth. Second, it assesses intra-layer compressibility by calculating the spectral entropy of projection modules, enabling fine-grained rank distribution. Finally, the authors propose an Iterative Residual Filling (IRF) mechanism to reclaim parameter gaps caused by integer-rank truncation, reallocating this leftover budget to critical modules. CGSVD outperforms some training-free baselines in perplexity and zero-shot reasoning tasks.

**Compliance With Llm Reviewing Policy:**

Affirmed.

**Final Justification:**

The author did not address all of my concerns, so I am maintaining my score.

**Key Questions For Authors:**

- Have you conducted any preliminary tests on larger architectures (e.g., LLaMA-3-70B)?

- Why were methods like SoLA, Basis Sharing, and Dobi-SVD excluded from the empirical evaluations?

- Could you please provide the Zero-shot accuracy and Perplexity metrics for the original, uncompressed models corresponding to Tables 1, 2, and 6? Given that SVD-based compression often incurs a heavy penalty on downstream tasks even at a 20-30% compression ratio, how do you justify the practical usability of this compression paradigm in real-world deployment scenarios if the absolute accuracy drop is excessively large?

- In Figure 2, you highlight inference speedup at 50% and 60% compression ratios. Given that the model's perplexity and reasoning capabilities have severely degraded at these levels (as shown in Table 1), what is the practical significance of reporting these throughput numbers? Are there specific use cases where a model with such low accuracy remains functionally useful?

**Limitations:**

yes

**Strengths And Weaknesses:**

### Strengths:

- The framework offers a well-motivated approach to rank allocation by combining macroscopic and microscopic metrics.

- The ablation studies clearly validate the proposed components. The results confirm that Layer Importance (LI), Module Compressibility (MC), and the IRF mechanism all independently and collectively contribute to mitigating perplexity degradation.

### Weaknesses:

- While the cascaded allocation strategy is novel, the foundational spatial reconstruction and whitening techniques heavily build upon existing activation-aware methods like ASVD and SVD-LLM.

- It remains an open question whether the observed layer-wise and module-wise redundancy profiles scale to much larger, frontier models (e.g., 70B+ parameters).

- Missing SOTA baselines like SoLA, Basis Sharing, and Dobi-SVD.

- Questionable practical utility.

- Performance on generative tasks like GSM8K (Math) and HumanEval (Code) is unclear.

Citations:

[1] Huang X, Huang Y L, Wen Z. Sola: Leveraging soft activation sparsity and low-rank decomposition for large language model compression. AAAI 2025.

[2] Wang J, Chen Y G, Lin I C, et al. Basis sharing: Cross-layer parameter sharing for large language model compression. ICLR 2025.

[3] Wang Q, Ke J, Tomizuka M, et al. Dobi-SVD: Differentiable SVD for LLM Compression and Some New Perspectives. ICLR 2025.

---

> ### Author Rebuttal · Authors · 2026-03-31
>
> Dear Reviewer oq3H:
>
> Thank you for taking the time to read and review our paper! In the following, we summarize your main concerns point by point.
>
> > Q1: The novelty of the proposed framework appears limited, as its spatial reconstruction and whitening design heavily build upon prior activation-aware SVD methods.
>
> Thank you for this important comment. We agree that the whitening-based reconstruction in CGSVD follows the standard activation-aware SVD pipeline and is not the primary novelty of our work. Instead, our main contribution is a general, training-free, non-uniform rank allocation framework that can be integrated with existing uniform-rank SVD methods. Specifically, CGSVD replaces the common “one-size-fits-all” rank assignment with a more cascaded and fine-grained strategy by jointly modeling inter-layer importance, intra-layer compressibility, and IRF-based budget refinement. This understanding is also aligned with other reviewers’ assessments: Reviewer sBiw highlighted that CGSVD improves over the prior “one size fits all” strategy through a “cascaded” allocation design, while Reviewer q7hw explicitly recognized that our method replaces uniform rank allocation with a two-level non-uniform policy. We will revise the paper to make this distinction clearer and better emphasize that the novelty of CGSVD lies in this broadly applicable allocation paradigm rather than in the whitening operation itself.
>
> > Q2: The scalability of the proposed layer-wise and module-wise allocation strategy to much larger frontier models remains unclear.
>
> Thank you for this important comment. We agree that evaluating on much larger frontier models would further strengthen the evidence for the scalability of CGSVD. However, due to hardware limitations, we are currently unable to conduct experiments on 70B+ models within the rebuttal period. Nevertheless, our current results already cover multiple representative architectures ranging from 3B to 13B and show consistent improvements over strong SVD-based baselines across different model families and scales. We agree that extending the evaluation to substantially larger models is an important future direction, and we will add this discussion more explicitly in the revised version.
>
> > Q3: Several relevant recent baselines, including SoLA, Basis Sharing, and Dobi-SVD, are missing from the empirical comparison.
>
> Thank you for this valuable comment. We would like to clarify that Basis Sharing has already been included in our additional comparison and is addressed in our response to Reviewer BKxC, Q5. Regarding Dobi-SVD, we exclude it because it is explicitly a training-based method, whereas our empirical comparison is restricted to training-free SVD compression methods, which is also the scope claimed by our paper. As for SoLA, we agree that it is a relevant recent method. However, due to the limited rebuttal period, we were not able to obtain stable reproduced results in time. We will clarify these scope considerations more explicitly in the revised version and, where possible, include a broader comparison in future updates.
>
> > Q4: The evaluation and practical utility are insufficiently justified, particularly due to the lack of generative-task results, original uncompressed-model metrics, and clearer evidence that the reported speedups remain meaningful under substantial performance degradation.
>
>  Thank you for this important comment. For completeness, we have included the perplexity and zero-shot accuracy of the original uncompressed models in the anonymous supplementary repository. Please refer to: `https://anonymous.4open.science/r/CGSVD-BD6E/result.md`. We have provided additional generative-task results, including GSM8K and HumanEval, in our response to **Reviewer q7hw, Q3**. Please refer to that response for the detailed results and discussion.
>
> We would like to clarify that the practical utility of SVD-based compression should be judged by the accuracy–efficiency trade-off, rather than by the absolute accuracy drop alone. In our view, the most relevant regime is moderate compression, where the model remains useful while already reducing memory and inference cost. Moreover, the compressed model is not necessarily the final endpoint, as it can be further combined with parameter-efficient fine-tuning to recover downstream capability.
>
> We also emphasize that the 50% and 60% compression ratios are reported mainly because they follow a similar evaluation setting to SVD-LLM, enabling a fair comparison under more challenging compression budgets. Our intention is not to suggest that such aggressive ratios are preferred for deployment. Rather, they are included to show the robustness frontier of different methods. In practice, we consider moderate compression ratios more meaningful for deployment, while higher ratios mainly serve as stress-test settings for method comparison. We will clarify this point in the revised version.

---

> > ### Author Rebuttal · Reviewer_oq3H · 2026-04-02
> >
> > The current response lacks specific examples. For instance, the statement 'We would like to clarify that the practical utility of SVD-based compression should be judged by the accuracy–efficiency trade-off' is somewhat vague; could you elaborate on this with more concrete details? Furthermore, there is a lack of supporting empirical evidence. Could you provide experimental results demonstrating whether the model's performance can be recovered to an acceptable level via post-training after moderate compression?
> >
> > To better defend the unique value of SVD, the authors could argue—and empirically demonstrate—that a hybrid approach of 'moderate SVD compression + moderate bit-width quantization' (e.g., 20% SVD combined with 4-bit quantization) yields superior performance compared to purely aggressive quantization (e.g., 2-bit quantization).
> >
> > Lastly, I suggest the authors take a closer comparison at AdaSVD.

---

> > > ### Author Response · Authors · 2026-04-04
> > >
> > > > Q1:To better defend the unique value of SVD, the authors could argue—and empirically demonstrate—that a hybrid approach of 'moderate SVD compression + moderate bit-width quantization' (e.g., 20% SVD combined with 4-bit quantization) yields superior performance compared to purely aggressive quantization (e.g., 2-bit quantization).
> > >
> > > Thank you for the helpful follow-up. We agree that our previous wording was too abstract, and we appreciate the opportunity to clarify it with more concrete evidence. To address your request for more direct empirical support, we additionally conduct a new experiment on LLaMA2-7B to compare aggressive quantization against a hybrid strategy of moderate SVD compression plus moderate quantization. Specifically, pure 2-bit quantization gives WikiText/C4/AVG = 1415.64/7826.66/31.68%, whereas 20% CGSVD + 4-bit gives 11.97/35.21/36.51%, and 20% CGSVD + LoRA + 4-bit further improves to 9.63/17.08/44.36%. In other words, compared with pure 2-bit quantization, the hybrid setting improves average zero-shot accuracy by 12.68 points after LoRA, while reducing perplexity by 1406.01 on WikiText and 7809.58 on C4. We believe this directly supports the unique practical value of SVD. We will add this comparison to the revised version.
> > >
> > > Table 1. Comparison between aggressive pure quantization and the proposed hybrid compression strategy on LLaMA2-7B. 2-bit denotes direct 2-bit quantization of the original model. 20% CGSVD + 4-bit denotes 20% compression followed by 4-bit quantization. 20% CGSVD + LoRA + 4-bit denotes a three-stage pipeline: we first apply 20% CGSVD structural compression to the original model, then perform LoRA-based post-training on the compressed model for performance recovery, and finally quantize the resulting model to 4-bit precision.
> > >
> > > | Method                  | WikiText |   C4 | OBQA | ARC_e | WinoG | HellaS | ARC_c | PIQA  | AVG   |
> > > |-------------------------|---------:|--------:|-----:|------:|------:|-------:|------:|------:|------:|
> > > | 2-bit                    | 1415.64  | 7826.66 | 14.4 | 25.25 | 49.57 | 25.43  | 24.66 | 50.76 | 31.68 |
> > > | 20% CGSVD + 4-bit        | 11.97    | 35.21   | 16.0 | 38.26 | 51.30 | 32.58  | 22.10 | 58.81 | 36.51 |
> > > | 20% CGSVD + LoRA + 4-bit | 9.63     | 17.08   | 20.8 | 50.04 | 56.51 | 42.13  | 29.18 | 67.46 | 44.36 |
> > >
> > > > Q2: I suggest the authors take a closer comparison at AdaSVD.
> > >
> > > Thank you for this helpful suggestion. We agree that AdaSVD is a relevant and important recent baseline for SVD-based LLM compression. At the same time, we would like to clarify an important reproducibility issue. Although the AdaSVD paper states that the code and models will be available and provides a GitHub link, the public repository currently still appears to be incomplete and marked with a “TODO / Complete this repository” notice, which makes strict reproduction and deeper controlled comparison difficult at this stage. We fully agree that a more comprehensive head-to-head comparison with AdaSVD would be valuable, and we will incorporate a more detailed experimental study once a complete and reproducible implementation becomes available.
> > >
> > > We sincerely thank the reviewer again for the thoughtful and constructive comments, which have greatly helped us improve the clarity and empirical support of this work. We also truly appreciate your careful evaluation and valuable suggestions. We wish you continued success in your research and all the best in your work.

---

### Official Review · Reviewer_q7hw · 2026-03-11

**Soundness:** 2
**Presentation:** 3
**Significance:** 3
**Originality:** 2
**Overall Recommendation:** 4
**Confidence:** 4

**Summary:**

This paper proposes CGSVD, a training-free LLM compression method built on activation-aware SVD. The core idea is that, keeping the SVD-LLM [1] style, e.g., whitening/decomposition, but replace the uniform rank allocation with a two-level non-uniform policy: layer budgets are adjusted using an angular-distance importance score, and module budgets are adjusted using spectral entropy, together with a final Iterative Residual Filling step tries to spend the leftover budget caused by integer rank rounding. The reconstructed layer is then implemented as two low-rank matrices after inverse whitening.

Empirically, the authors report gains over standard SVD, and SVD-based compression methods (e.g., SVD-LLM, ASVD...) across several Llama-family and Vicuna model.

**Compliance With Llm Reviewing Policy:**

Affirmed.

**Final Justification:**

I believe authors addressed my concern in the rebuttal. Therefore, I increase the overall score.

**Key Questions For Authors:**

**Q1.** How robust is the layer-importance signal? Please show whether the angular-distance score is stable across different calibration sets, prompt distributions, and sequence lengths, and whether using the last token is materially better than averaging over tokens.

**Q2.** What exactly is the LoRA fine tuning protocol? The current discussion is too brief. Please specify data, task, training steps, rank, learning rate, and whether all methods are tuned identically.

**Limitations:**

Yes

**Strengths And Weaknesses:**

**S1.** Overall, this paper targets an important and practically problem: training-free compression of LLMs with hardware-friendly low-rank structure rather than unstructured sparsity. This problem is worth studying, and the paper positions itself clearly against low-rank SVD approaches such as ASVD, SVD-LLM and so forth. The main argument is also easy to follow: rank should not be allocated uniformly across depth and module. The reported improvements over SVD-based methods are fairly consistent, and the ablations supports that each component contributes, especially can explain the higher-compression regime.

**S2.** I also think the method is directionally sensible: using representation change to score layer sensitivity and singular spectrum shape to score module compressibility is a reasonable design choice for a training-free method. The overall architecture is straightforward and implementable, and the reported compression cost is much smaller than baselines, like ASVD. These give this work practical value.

**W1.** My main issue is that, this paper's central allocation rules, although straightforward and seemingly effective, are quite heuristic. Angular distance on the last token only is used as the layer-importance proxy, but the paper does not convincingly show that this is stable across prompts, sequence lengths, or calibration sets, nor does it compare against simpler alternatives such as, for example,  full-sequence averaging, activation norm change, or reconstruction error based sensitivity. Similarly, spectral entropy is unclear to me, but the paper does not really establish that it is the right compressibility signal beyond one ablation against a uniform baseline.

**W2.** The evaluation is decent but not yet strong enough for the paper's claim. The main results cover perplexity and zero-shot, which is useful, but still narrow for broad claims about preserving the semantic expressive power or reasoning capabilities. I would like to see broader evaluation on, for example, math reasoning GSM8K, and please provide more detail on your LoRA setup.

---

> ### Author Rebuttal · Authors · 2026-03-31
>
> Dear Reviewer q7hw:
>
> Thank you for taking the time to read and review our paper! In the following, we summarize your main concerns point by point.
>
> > Q1: The robustness of the layer-importance signal, including the use of angular distance on the last token, is not sufficiently validated.
>
> Thank you for this important suggestion. We have provided a detailed analysis of the stability of the layer-importance signal across different calibration settings in our response to Reviewer sBiw, Q4, and we kindly refer the reviewer to that discussion for brevity. In addition, following your suggestion, we further compare our angular-distance based design with several alternative layer-importance criteria, including token-averaged angular distance, activation norm change, reconstruction error, and cosine similarity. The results show that our method achieves the best performance among these alternatives, which further supports the effectiveness of using angular distance on the last token as the layer-importance signal. We will add this comparison and discussion to the revised version.
>
> Table 1. Comparison of different layer-importance criteria on 30% LLaMA2-7B.
>
> | Method | angular_avg | norm_change | recon_error | cosine | Ours |
> |---|---:|---:|---:|---:|---:|
> | PPL | 9.41 | 9.99 | 10.32 | 9.40 | 9.37 |
>
> > Q2: The choice of spectral entropy as the module compressibility criterion needs stronger justification.
>
> Response: Thank you for this helpful comment. We would like to clarify that the effectiveness of spectral entropy is already supported by the ablation study in the paper, where the module compressibility (MC) component consistently improves performance over the uniform allocation baseline across different compression ratios. This indicates that introducing a module-level compressibility signal is beneficial and that spectral entropy provides an effective criterion for non-uniform rank allocation. In addition, following a related suggestion from Reviewer BKxC (Q4), we further compare spectral entropy with alternative criteria, and the results also support the superiority of our current design.
>
> > Q3: The evaluation is still limited and should be extended to broader reasoning benchmarks.
>
> Thank you for this helpful suggestion. We further extend the evaluation to broader reasoning benchmarks by testing the 30% compressed LLaMA3.1-8B-Instruct model after fine-tuning on HumanEval and GSM8K. The results show that our method consistently outperforms SVD-LLM on both benchmarks, indicating that CGSVD better preserves the model’s reasoning and code-generation capabilities beyond perplexity and zero-shot commonsense evaluation. We will add these results to the revised version.
>
> Table 2. Fine-tuning results on broader reasoning benchmarks for 30% compressed LLaMA3.1-8B-Instruct.
>
> | Method  | HumanEval | GSM8K |
> |---------|----------:|------:|
> | SVD-LLM | 40     | 10 |
> | Ours    | 55    | 12.58 |
>
> > Q4: The LoRA fine-tuning setup needs to be described more clearly and completely.
>
> Thank you for this helpful suggestion. We agree that the LoRA fine-tuning protocol should be described more clearly. In the revised version, we will explicitly provide the detailed setup, including the dataset, training hyperparameters, and LoRA configuration used in our experiments. Specifically, the fine-tuning is conducted on the Alpaca dataset, with a batch size of 128, micro batch size of 4, 2 epochs, learning rate of 1e-5, LoRA rank of 8, LoRA alpha of 16, dropout of 0.05, and maximum sequence length of 256, using a validation set of size 2000. We will also clarify that the compared methods are evaluated under the same LoRA setting to ensure fairness.

---

> > ### Author Rebuttal · Reviewer_q7hw · 2026-04-03
> >
> > Thanks for your rebuttal. I think my concerns have been addressed. Good luck.

---

> > > ### Author Response · Authors · 2026-04-03
> > >
> > > Thank you for your time and for carefully reading and considering our response. We sincerely appreciate your positive recognition of our explanation.

---

### Official Review · Reviewer_BKxC · 2026-03-11

**Soundness:** 3
**Presentation:** 3
**Significance:** 2
**Originality:** 2
**Overall Recommendation:** 3
**Confidence:** 4

**Summary:**

This paper proposes CGSVD, an SVD-based compression method for large language models. Unlike previous SVD-based approaches that apply uniform compression across the model, CGSVD allocates different compression ratios to different layers and modules according to their importance. This fine-grained allocation enables the model to achieve better performance under a given compression budget.

**Compliance With Llm Reviewing Policy:**

Affirmed.

**Key Questions For Authors:**

Key Questions for authors:
1.	In the clip formular, how are \rho_min and \rho_max are determined? Do they keeps the same for the whole model? Any insights behind it?
2.	It is not clear for me why entropy is used to evaluate the module compressibility. Why can not use varience or something else?
3.	The authors should compared their method with the approach proposed in [1], which addresses a similar problem.
[1] "Basis Sharing: Cross-layer Parameter Sharing for Large Language Model Compression" [Wang et al., 2024]

**Limitations:**

see the weakness and questions.

**Strengths And Weaknesses:**

Strengths:
This paper is easy to understand and the training free rank allocation methods outperforms the previous SVD-based methods.
Weakness:
1.	The main results are based on LlaMA2-7B which is out of date, and all the results are based on base models. It is not clear whether this method can be applied on some new released models like qwen3 or some reasoning or instruction fine tuned models.
2.	The inter layer rank allocation highly depends on the calibhration dataset, but the the impact of calibration dataset selection is not clearly analyzed.

---

> ### Author Rebuttal · Authors · 2026-03-30
>
> Dear Reviewer BKxC:
>
> Thank you for taking the time to read and review our paper! In the following, we summarize your main concerns point by point.
>
> > Q1: The generalizability of the method to newer and instruction-/reasoning-tuned models remains unclear.
>
> Thank you for this valuable suggestion. We further evaluate CGSVD on newer instruction-tuned models under the same compression setting, and the results consistently show that our method remains more effective than the baseline. We will include these results and the corresponding discussion in the revised version.
>
> Table 1. Performance on instruction-tuned LLaMA models at a 30% compression ratio.
> | Model | Method | WikiText2 | C4 | OBQA | ARC_e | WinoG | HellaS | ARC_c | PIQA | AVG |
> |---|---|---:|---:|---:|---:|---:|---:|---:|---:|---:|
> | LLaMA3.1-8B-Instruct | SVD-LLM | 124.17 | 267.06 | 12.80 | 26.73 | 51.22 | 25.69 | 21.93 | 52.61 | 31.83 |
> | LLaMA3.1-8B-Instruct | Ours | 99.25 | 213.27 | 11.20 | 39.39 | 52.01 | 28.82 | 18.94 | 58.60 | 34.83 |
> | LLaMA3.2-3B-Instruct | SVD-LLM | 208.83 | 439.14 | 11.80 | 25.25 | 51.54 | 25.93 | 22.27 | 51.58 | 31.40 |
> | LLaMA3.2-3B-Instruct | Ours | 156.34 | 286.14 | 12.20 | 33.84 | 51.78 | 28.79 | 20.48 | 57.56 | 34.11 |
>
> > Q2: The effect of calibration dataset selection is not sufficiently analyzed.
>
> Thank you for this important comment. We have provided a detailed analysis of the effect of calibration dataset selection and calibration settings in our response to Reviewer sBiw, Q4. For brevity and to avoid repetition, we kindly refer the reviewer to that response, where additional experiments and discussion are provided.
>
> > Q3: The choice of (\rho_{\min}) and (\rho_{\max}) needs clarification.
>
> Thank you for this question. In our current implementation, we set $\rho_{\min}$=0.05 and $\rho_{\max}$=0.95 as fixed bounds for the entire model. These two hyperparameters are introduced to constrain the clipped retention ratios in both layer-wise and module-wise allocation, so that the assigned budgets can adapt to importance differences while remaining within a reasonable range. In practice, this prevents certain layers or modules from being allocated an excessively small retention ratio that may severely harm their functional capacity, as well as an excessively large ratio that would weaken the intended compression effect. We adopt fixed values mainly for stability and simplicity across different compression settings, allowing the proposed allocation strategy to flexibly adjust to different target compression ratios without introducing additional hyperparameter complexity. We will clarify this design choice more explicitly in the revised version.
>
> > Q4: The use of entropy for compressibility assessment needs better justification.
>
> Thank you for this insightful suggestion. We use spectral entropy because it is defined on the normalized singular-value energy distribution and better reflects how information is distributed across latent dimensions. In our setting, module compressibility depends on whether the spectral energy is concentrated in a few dominant components or spread across many dimensions. Spectral entropy aligns well with this objective: low entropy indicates stronger low-rank compressibility, while high entropy suggests that the module is harder to approximate at low rank. By contrast, variance mainly measures magnitude dispersion and does not explicitly capture this global energy distribution. Our additional comparison also supports the suitability of spectral entropy, and we will add this clarification to the revised version.
>
> Table 2. Comparison between the variance-based alternative and our entropy-based compressibility criterion on LLaMA2-7B.
>
> | Method   | 30%    | 40%    | 50%    |
> |----------|-------:|-------:|-------:|
> | Variance | 5994.43 | 6869.11 | 9988.90 |
> | Ours     | 9.37    | 12.74   | 21.92   |
>
> > Q5: A comparison with Basis Sharing is needed.
>
> Thank you for this valuable suggestion. Following your recommendation, we further compare our method with Basis Sharing under compression ratios from 30% to 60%. The results show that our method consistently achieves the best performance across all settings. In particular, while Basis Sharing improves over SVD-LLM, our method further reduces perplexity at every compression ratio, and the advantage becomes more pronounced at higher compression levels. These results provide additional evidence that the proposed non-uniform rank allocation strategy is more effective for preserving model quality under a fixed compression budget. We will add this comparison to the revised version.
>
> Table 3. Perplexity comparison with Basis Sharing under different compression ratios on LLaMA2-7B.
>
> | Method | 30% | 40% | 50% | 60% |
> |---|---:|---:|---:|---:|
> | SVD-LLM | 10.66 | 16.14 | 33.29 | 89.95 |
> | Basis Sharing | 9.69 | 13.87 | 23.77 | 50.38 |
> | Ours | 9.37 | 12.74 | 21.92 | 26.69 |

---

### Official Review · Reviewer_sBiw · 2026-03-23

**Soundness:** 3
**Presentation:** 4
**Significance:** 3
**Originality:** 3
**Overall Recommendation:** 5
**Confidence:** 4

**Summary:**

CGSVD is a training-free LLM compression framework that replaces standard uniform SVD with a dual-level, non-uniform rank allocation strategy. It identifies critical parameters by measuring inter-layer significance via angular distance and intra-layer compressibility through spectral entropy. A specialized Iterative Residual Filling (IRF) mechanism further optimizes the parameter budget after truncation. Validated on LLaMA and Mistral models, CGSVD significantly outperforms uniform SVD baselines in both zero-shot accuracy and perplexity.

**Compliance With Llm Reviewing Policy:**

Affirmed.

**Final Justification:**

Thank you for the detailed rebuttal. I particularly appreciate the new throughput data in Table 1, which provides the empirical evidence needed to support your hardware efficiency claims. The sensitivity analyses in Tables 2 and 3 also clearly demonstrate the robustness of your allocation strategy.

While I understand the hardware constraints regarding 70B+ models and fine-tuning comparisons, I encourage you to include these resource-based justifications in the final manuscript to clearly define the method's current operational bounds. Overall, your responses have addressed my primary concerns. I am satisfied with the clarifications and will maintain my recommendation for Acceptance.

**Key Questions For Authors:**

- I am interested in understanding how sensitive the rank allocation is to the choice of calibration data, as the method relies on a calibration set. How would a domain-specific calibration set bias the compressed model?
- "hardware-friendly pathway": Can authors conduct a latency benchmark to demonstrate the actual throughput gains compared to the baseline?

**Limitations:**

Yes

**Strengths And Weaknesses:**

## Strengths

- Unlike prior SVD methods that treat every layer or module identically, CGSVD applies a "cascaded" approach to rank allocation. This is better than the "one size fits all" approach covered in previous literature.

- The paper utilizes spectral entropy to measure the intrinsic compressibility of specific weight matrices. Previously, people have used energy to measure the information present. This is an interesting difference.

- Use of angular distance over cosine similarity for identifying layer sensitivity is mathematically justified.

- Some other feats that I like are training free framework, good use of Iterative Residual Filling, thorough empirical evidence (a 6.08% zero-shot accuracy boost and a 33.39 reduction in perplexity).


## Weaknesses

- I believe even though the method is training-free, the pre-computation step is complicated and can be resource-intensive. I would like to know how it changes with the model size (e.g., a 70B-parameter model).

- Experiments are conducted on the model up to a size of 13B parameters. It would be nice if authors could include a large model (100B+) if possible to showcase that the method is generalizable.

- Can we get a comparison of how inference speed varies with the different methods (different compression ratios/#parameters)? As the number of operations is increasing, even though the absolute #parameters is decreasing.

- Can we also discuss the comparison between this "training free" method vs. a small, super light weight finetuning of the model? How the performance will change assuming the same time for both the methods?

---

> ### Author Rebuttal · Authors · 2026-03-30
>
> Dear Reviewer sBiw:
>
> Thank you for taking the time to read and review our paper! In the following, we summarize your main concerns point by point.
>
> > Q1: The pre-computation cost, scalability, and generalizability of the method on larger models remain unclear.
>
> Thank you for this valuable suggestion. We agree that evaluation on larger models would further strengthen the evidence for the scalability and generalizability of our method. However, due to hardware limitations, we are currently unable to conduct experiments on models larger than 13B within the rebuttal period. Our present experimental platform cannot reliably support compression and evaluation at substantially larger scales under the available resource budget. Nevertheless, our current results already cover multiple representative LLM families across the 3B–13B range and consistently demonstrate the effectiveness of the proposed method on different architectures and model sizes. We fully acknowledge the importance of extending the evaluation to larger models, and we will pursue this direction in future work when additional computational resources become available.
>
> > Q2: More comprehensive deployment-oriented evaluation, including inference speed comparisons across methods and compression ratios.
>
> Thank you for this valuable suggestion. We agree that practical efficiency should be evaluated by actual inference performance rather than parameter reduction alone. In response, we further compared the inference speed of our method and SVD-LLM under multiple compression ratios. The results show that our method consistently delivers higher throughput across all tested settings, indicating that its benefit extends beyond improved perplexity and zero-shot performance to tangible deployment efficiency. We will incorporate this comparison into the revised manuscript to further substantiate the hardware-friendly advantage of our method.
>
> Table 1. Inference throughput comparison of SVD-LLM and CGSVD across different compression
> | Ratio | SVD-LLM  | Ours  |
> |-:|-:|-:|
> | 30%   | 60.94              | 79.57          |
> | 40%   | 68.35              | 87.51          |
> | 50%   | 74.72              | 96.37          |
> | 60%   | 81.12              | 104.74         |
>
> > Q3: A comparison with lightweight fine-tuning methods under a similar time/computation budget would be valuable.
>
> Thank you for this insightful suggestion. We agree that comparing our method with lightweight fine-tuning based approaches would be valuable for a more comprehensive discussion. However, even so-called lightweight fine-tuning methods generally require substantially more computational resources than a training-free pipeline, since they still involve gradient updates, optimizer states, and multiple training iterations. In our current setting, limited to two NVIDIA A40 GPUs, we are unable to reliably run methods such as DobiSVD and ARS even on 7B-scale models within the rebuttal period. For this reason, we focus on training-free baselines that are feasible under the same hardware constraints. We will include a clearer discussion of this practical trade-off in the revision and consider such comparisons in future work when additional computing resources are available.
>
> > Q4: The sensitivity of rank allocation to the calibration data, especially domain-specific data, needs clarification.
>
> Thank you for this important question. Our additional results show that CGSVD is not highly sensitive to the calibration setting, although richer calibration data does improve rank allocation quality. On 30% LLaMA2-7B, increasing the batch size from 16 to 512 reduces perplexity from 11.31 to 9.29 on WikiText2 and from 19.22 to 14.93 on C4; increasing the sequence length from 128 to 4096 further reduces perplexity from 20.28 to 8.65 and from 34.67 to 14.25, respectively. These results suggest that CGSVD is generally robust to calibration settings, while domain-specific calibration data may still introduce some distribution bias. We will add this clarification and the corresponding experimental results to the revised version.
>
> Table 2. Sensitivity of CGSVD to calibration batch size on 30% LLaMA2-7B.
>
> | Batch Size | 16 | 32 | 64 | 128 | 256 | 512 |
> |---|---:|---:|---:|---:|---:|---:|
> | WikiText2 | 11.31 | 10.26 | 9.79 | 9.54 | 9.37 | 9.29 |
> | C4 | 19.22 | 16.87 | 15.84 | 15.65 | 15.13 | 14.93 |
>
> Table 3. Sensitivity of CGSVD to calibration sequence length on 30% LLaMA2-7B.
>
> | Seq\_len | 128 | 256 | 512 | 1024 | 2048 | 4096 |
> |---|---:|---:|---:|---:|---:|---:|
> | WikiText2 | 20.28 | 15.18 | 12.18 | 10.54 | 9.37 | 8.65 |
> | C4 | 34.67 | 25.18 | 20.40 | 17.77 | 15.13 | 14.25 |

---

> > ### Author Rebuttal · Reviewer_sBiw · 2026-04-05
> >
> > Thank you for the detailed rebuttal. I particularly appreciate the new throughput data in Table 1, which provides the empirical evidence needed to support your hardware efficiency claims. The sensitivity analyses in Tables 2 and 3 also clearly demonstrate the robustness of your allocation strategy.
> >
> > While I understand the hardware constraints regarding 70B+ models and fine-tuning comparisons, I encourage you to include these resource-based justifications in the final manuscript to clearly define the method's current operational bounds. Overall, your responses have addressed my primary concerns. I am satisfied with the clarifications and will maintain my recommendation for Acceptance.

---

> > > ### Author Response · Authors · 2026-04-06
> > >
> > > Thank you very much for your positive and encouraging feedback. We sincerely appreciate your recognition of the new throughput results in Table 1 and the sensitivity analyses in Tables 2 and 3. We are glad that these additional experiments helped clarify the hardware efficiency and robustness of our method.
> > >
> > > We also thank you for the helpful suggestion regarding the operational bounds of the method. In the final manuscript, we will explicitly include the resource-based justifications for not extending the evaluation to 70B+ models and fine-tuning comparisons, so as to clearly define the current practical scope and limitations of our approach. If computational resources permit, we will also include these additional evaluations in the revised version.
> > >
> > > We greatly appreciate your time, careful evaluation, and your continued support of our work. We wish you all the best and thank you again for your valuable comments.

---

### Decision · Program_Chairs · 2026-04-30

**Decision:**

Accept (regular)

**Comment:**

Authors of this work propose CGSVD, a training free, SVD-based LLM compression framework that replaces uniform rank allocation with a cascaded, dual-level non-uniform policy wherein the layers are ranked by angular distance base importance score and modules are ranked by spectral entropy. The method additionally adds an iterative residual filling step to redistribute residual ranks. The method yields consistent improvement on LLaMa family of models in the range 3-13B.

The reviewers consistently agree about the importance of the problem this work is trying to solve and acknowledge the combining of dual level metrics in compression is a meaningful contribution beyond uniform rank allocation. The ablation studies further demonstrate each components contribution to the improvements.

Despite the support for the contributions, several concerns regarding the work remained which include:
- The paper can improve its empirical comparison by including recent non-uniform and/or training-based baselines such as AdaSVD, SVD-LLM-v2, etc.). The authors did provide additional baselines during the rebuttal period include those against Basis Sharing and SVD-LLM which was welcomed by most reviewers.
- Some reviewers still hold the concern that the work might not garner a strong adoption given these missing comparisons for practitioners to make informed choices about choosing this method.

Given the importance of the problem within the training-free regime, I believe that CGSVD provides a clear technical contribution to the domain backed by comprehensive experiments and ablations confirming its effectiveness towards gains as well as hardware speedups while maintaining robustness. I would request the authors to further improve their draft by considering the addition of comparison against other methods mentioned by the reviewers particularly AdaSVD and SVD-LLM-v2 or at least adding a discussion about this consideration.